



# A WRF-Chem study of the greenhouse gas column and in situ surface concentrations observed at Xianghe, China. Part 1: Methane (CH$_4$)

Sieglinde Callewaert[1,3], Minqiang Zhou[1,2], Bavo Langerock[1], Pucai Wang[2], Ting Wang[2], Emmanuel Mahieu[3], and Martine De Mazière[1]

[1]Royal Belgian Institute for Spacy Aeronomy (BIRA-IASB), Brussels, Belgium
[2]CNRC & LAGEO, Institute of Atmospheric Physics, Chinese Academy of Sciences, Beijing, China
[3]UR SPHERES, Department of Astrophysics, Geophysics and Oceanography, University of Liège, Liège, Belgium

**Correspondence:** Sieglinde Callewaert (sieglinde.callewaert@aeronomie.be), Minqiang Zhou (minqiang.zhou@aeronomie.be)

**Abstract.** This study is the first of two companion papers which investigate the temporal variability of $CO_2$, $CH_4$ and additionally CO concentrations measured at the Xianghe observation site near Beijing in China using the Weather Research and Forecast model coupled with Chemistry (WRF-Chem), aiming to understand the contributions from different emission sectors and the influence of meteorological processes. Simulations of the in situ (PICARRO) and remote sensing (TCCON-affiliated)

measurements are produced by the model's greenhouse gas option, called WRF-GHG, from September 2018 until September 2019. The present study discusses the results for $CH_4$. The model shows good performance, after correcting for biases in boundary conditions, achieving correlation coefficients up to 0.66 for near-surface concentrations and 0.65 for column-averaged data. The simulations use separate tracers for different source sectors and revealed that energy, residential heating, waste management and agriculture are the primary contributors to the $CH_4$ concentrations, with the energy sector having a greater impact on

column measurements than surface concentrations. Monthly variability is linked to both emission patterns and meteorological influences, with advection of either clean or polluted air masses from the North China Plain playing a significant role. The diurnal variation of the in situ concentrations due to planetary boundary layer dynamics is quite well captured by WRF-GHG. Despite capturing the key variability of the $CH_4$ observations, the model displays a seasonal bias, likely originating from an incorrect seasonality in the emissions from agricultural and/or waste management activities. Our findings highlight the value

of WRF-GHG to interpret both surface and column observations at Xianghe, offering source sector attribution and insights in the link with local and large-scale winds based on the simultaneously computed meteorological fields. However, they also highlight the need to improve the knowledge on the seasonal $CH_4$ cycle in northern China to obtain more accurate emission data and boundary conditions for high-resolution modeling.

## 1 Introduction

Carbon dioxide ($CO_2$) and methane ($CH_4$) are the most important anthropogenic greenhouse gases (GHG), contributing to climate change. Driven by human activities, the atmospheric burden of both species has been increasing over the last 200 years



to unprecedented levels (Masson-Delmotte et al., 2021). Moreover, $CH_4$ has a 28 times larger global warming potential than $CO_2$ over a period of 100 year and a 10 times shorter atmospheric lifetime. Controlling $CH_4$ emissions is therefore a priority to mitigate climate change in the near future (Saunois et al., 2020).

Because of rapid industrialization in the past decades and its heavy dependence on coal, China is the world's largest emitter of $CO_2$ and $CH_4$ (Friedlingstein et al., 2022; Worden et al., 2022). The main anthropogenic $CO_2$ sources in China are industry, power generation, residential and commercial activities and transportation (Zhao et al., 2012), while sectors such as coal mining, livestock, rice paddies, landfills and wastewater management are the largest contributors to the $CH_4$ emissions in China (Chen et al., 2022). China has pledged to reach its carbon peak by 2030 and neutrality by 2060. To help battle climate change
and reach these goals, it is essential to have accurate observations of the GHG concentrations. Not only does atmospheric monitoring aid in revealing sources and sinks and controlling the impact of mitigation measures, but by studying temporal variations a better understanding of the carbon cycle and its interactions with the atmosphere can be achieved.

Since 2018, both ground-based in situ and remote sensing observations of GHGs have been deployed at the Xianghe observatory, which is located about 50 km southwest of Beijing. Its location in the center of the Beijing-Tianjin-Hebei (BTH)
megalopolis makes it an interesting site to study the properties and variability of GHGs in a polluted area. The remote sensing observations are made by a Fourier Transform Infrared (FTIR) spectrometer and are part of the international Total Column Carbon Observing Network (TCCON), while the in situ concentrations are measured by a PICARRO cavity ring-down spectroscopy (CRDS) analyzer that samples air from a tower at an altitude of 60 m above the ground.

Our work aims to perform a comprehensive analysis of both in situ and column observations of $CO_2$, $CH_4$ and additionally CO
at Xianghe to gain a better understanding of the causes of the observed temporal variabilities and complement previous studies. The present article is the first of two companion papers where the focus of the current work lies on the $CH_4$ observations. A second paper (in preparation) will cover the analysis for $CO_2$ and CO.

Some first insights in the observed $CH_4$ time series at Xianghe were made by Yang et al. (2020) and Ji et al. (2020). They found that the seasonal cycle of $XCH_4$ is different compared to those at other TCCON sites at similar latitude, with larger
concentrations in summer and autumn and lower values in spring. Furthermore, the column observations of $CO_2$, $CH_4$ and CO show a large day-to-day variability and are correlated with each other. Yang et al. (2020) showed that the high values are related to both local pollution and pollution originating from the south, while low concentrations are corresponding with clean airmasses from more remote regions in the north.

To achieve our goal, we will simulate the time series at a high spatial resolution with the WRF-Chem model for greenhouse
gases (WRF-GHG). This widely used regional atmospheric transport model simulates the 3-D concentrations together with meteorological fields without chemical interactions, which is generally a valid assumption regarding the regional domain and the relatively long atmospheric lifetimes of the target species ($\sim$ 100 yrs for $CO_2$, $\sim$ 10 yrs for $CH_4$ and several weeks for CO)(Dekker et al., 2017). Nevertheless, both $CH_4$ and CO are prone to chemical reactions in the atmosphere, making this assumption a simplification of actual conditions, which should be taken into account when analyzing the results. WRF-GHG
has already shown to be a useful tool to study $CO_2$ fluxes and variability in China (Dayalu et al., 2018; Liu et al., 2018; Li et al., 2020; Dong et al., 2021). However, and to our best knowledge, applications to $CH_4$ or CO observations in China have



not been reported yet. Elsewhere, this model was successfully used to analyze comparable observations (Zhao et al., 2019; Hu et al., 2020; Park et al., 2020; Callewaert et al., 2022). Therefore, this study will additionally assess the model's capability of simulating these time series in north China and highlight its strengths and weaknesses in this region.

This work is structured as follows: in Sect. 2 the Xianghe site and its observations are described, together with the XCH$_4$ product of TROPOMI (the TROPOspheric Monitoring Instrument onboard Sentinel-5P), which will give additional insight into the results. Further, an overview of the WRF-GHG model system is given and the approach used to compare the model simulations with the different measurements. Section 3 presents the results and discussion: the main model performance is evaluated in Sect.3.1, followed by an analysis of the contributions from different source sectors to the CH$_4$ observations at

Xianghe in Sect.3.2. Section 3.3 explores potential causes of the observed seasonal bias in the model simulations, while Sect. 3.4 examines the key meteorological processes influencing CH$_4$ variability. Further, a comparison with TROPOMI XCH$_4$ is conducted in Sect. 3.5 to investigate the potential overestimation of emissions from a coal mine source near Tangshan. Finally, Sect.4 summarizes the key findings and conclusions for CH$_4$.

## 2   Data, models and methods

### 2.1   Xianghe site


The observation site is situated in Xianghe county (39.7536° N, 116.96155° E; 30 m a.s.l.), a suburban area in the Beijing-Tianjin-Hebei (BTH) region in north China. The center of Xianghe is about 2 km to the east of the site, while the metropolitan cities of Beijing and Tianjin are located about 50 km to the northwest and 70 km to the south-southeast, respectively (see Fig. 1b). Cropland and irrigated cropland are the predominant kind of vegetation in the area. The East Asian Monsoon, which

causes hot, humid summers with plenty of precipitation and cold, dry winters, determines the climate.

Since 1974, atmospheric observations are made at the Xianghe observatory by the Institute of Atmospheric Physics (IAP), Chinese Academy of Sciences (CAS). In June 2016 a FTIR spectroscopy instrument (Bruker IFS 125HR) was installed on the roof of the observatory, two years later, a solar tracker was added to the setup and continuous measurements are made from June 2018 onwards. This ground-based remote sensing instrument measures spectra in the infrared and is affiliated with

TCCON (Wunch et al., 2011; Zhou et al., 2022), providing total column-averaged dry air mole fractions (denoted as Xgas) of CO$_2$, CH$_4$ and CO. In the current study, the GGG2020 data version (Laughner et al., 2024) is used. Depending on the weather and measurement status, observations occur every 5-20 min. TCCON measurements are performed under clear sky conditions only. The measurement uncertainty is about 6 ppb for XCH$_4$. Further details about the instrument and retrieval methodology can be found in Yang et al. (2020).

Additionally, in situ mole fractions of CO$_2$ and CH$_4$ are measured by a PICARRO cavity ring-down spectroscopy G2301 analyzer since June 2018. The instrument samples air from an inlet fixed at 60 m above the ground on a tower. More detail about the measurement setup is given in Yang et al. (2021). The measurement uncertainty is 1 ppb for CH$_4$.



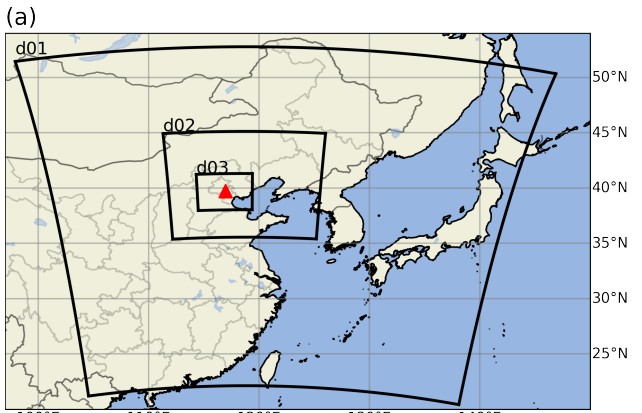
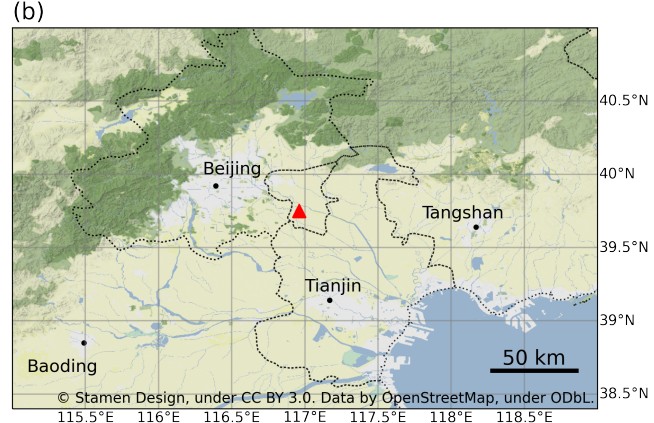

**Figure 1.** (a) Location of the WRF-GHG domains, with horizontal resolutions of 27 km (d01), 9 km (d02) and 3 km (d03). All domains have 60 (hybrid) vertical levels extending from the surface up to 50 hPa. (b) Terrain map including the largest cities in the region of Xianghe, roughly corresponding to d03. The location of the Xianghe site is indicated by the red triangle in both maps.

## 2.2 TROPOMI

The TROPOMI instrument on board the Sentinel-5 Precursor (S5P) satellite is observing the Earth on a polar sun-synchronous orbit. With a daily global coverage, it measures solar backscatter in the near and shortwave infrared absorption bands of which column-average mixing ratios of $CH_4$ can be retrieved. In the current study, the bias-corrected reprocessed L2 RemoTec-S5P $XCH_4$ product from SRON (ESA, 2021) was used, where a quality filter of 1.0 was applied. This L2 product was evaluated at Xianghe by Yang et al. (2020) and Tian et al. (2022): they found a small negative bias of -0.6% and -0.39% with TCCON $XCH_4$, respectively. These values are well within the mission requirements of 1.5 % and therefore indicate a good quality of TROPOMI $XCH_4$ in this part of China.

## 2.3 WRF-GHG modelling system

We use the Weather Research and Forecasting model coupled with Chemistry version 4.1.5 (WRF-Chem, Grell et al. (2005); Skamarock et al. (2019); Fast et al. (2006)) in its greenhouse gas option, called WRF-GHG (Beck et al., 2011). WRF-GHG is a Eulerian atmospheric transport model that simulates the 3-D concentration of trace gases at every time step simultaneously with meteorological fields, neglecting chemical reactions. The model configuration consists of three nested domains with increasing resolution in a Lambert Conformal projection (see Fig. 1a). The parent domain (d01) has 134 by 130 grid cells of $27 \times 27$ km$^2$ and covers a large part of China, Mongolia, North and South Korea and Japan. The second domain (d02), which has 133 by 121 grid cells of $9 \times 9$ km$^2$, mainly covers north China. Finally, the innermost domain (d03) has a resolution of $3 \times 3$ km$^2$ over 145 by 124 grid cells and almost completely covers BTH. There are 60 vertical levels between the surface and 50 hPa. A set of physical parameterization schemes was chosen (see Table 1) after performing several sensitivity tests which are detailed in Appendix A. Given the wide range of global anthropogenic emission datasets available and the significance of these fluxes



| Physics | Scheme name | Option |
|---|---|---|
| Microphysics | Morrison 2-moment | 10 |
| Longwave radiation | RRTMG | 4 |
| Shortwave radiation | RRTMG | 4 |
| Planetary boundary layer | Mellor-Yamada-Janjic | 2 |
| Surface layer | Eta similarity | 2 |
| Cumulus | Grell 3D Ensemble | 5 |
| Land surface | Unified Noah Land Surface Model | 2 |

**Table 1.** Overview of physical parameterization options used for WRF-GHG simulations.

to simulate accurate concentrations in regions with large anthropogenic activity such as BTH, several anthropogenic emission inventories were also included in these sensitivity tests.

### 2.3.1 Input data and parameterization

The model was driven by the hourly European Centre for Medium-Range Weather Forecasts (ECMWF) global ERA5 reanalysis data set ($0.25° \times 0.25°$, Hersbach et al. (2023a, b)) for meteorological fields. The concentration fields for $CO_2$ and $CH_4$ are initialized by the 3-hourly Copernicus Atmosphere Monitoring Service (CAMS) global reanalysis for greenhouse gases (EGG4), while the 6-hourly reactive gases product is used for CO (EAC4, Inness et al. (2019)). These CAMS reanalysis data sets are also used at the model domain boundaries to represent influences coming from outside the parent domain (d01). The

evolution of these initial and lateral boundary conditions inside the domain over time is stored in a separate tracer, the so-called background tracer. Similarly, the evolution of concentrations caused by emissions within the boundaries of d01 is saved in different tracers, dependent on their source sector. The sum of all tracers, including the background, gives the total simulated concentrations which can be compared to the observations.

The simulations are re-initialized with the ECMWF ERA5 data every 30 h, starting at 18:00 UTC the previous day with a 6 h

spin-up period, as done in other WRF-GHG modelling studies (Feng et al., 2016; Park et al., 2018; Pillai et al., 2011). Every day at 00:00 UTC, the tracer fields from the previous run are copied to the new simulation to ensure continuous transport of the concentrations.

We conducted sensitivity tests to identify a set of physical parameterization schemes and anthropogenic fluxes that provide appropriate simulations for all three species ($CH_4$, $CO_2$, and CO) across the different observation methods (in situ and remote

sensing). The details of these tests are provided in Appendix A. Our findings indicate that the anthropogenic fluxes from CAMS-GLOB-ANT v5.3 (Granier et al., 2019; Soulie et al., 2023) for $CO_2$ and $CH_4$, and from REAS v3.2.1 (Regional Emission Inventory in Asia, Kurokawa and Ohara (2020)) for CO offer the best alignment with the Xianghe observations. We released all fluxes in the lowest model layer near the surface and multiplied them with temporal factors of CAMS-TEMPO (Guevara et al., 2021) to account for hourly and daily variation. Remark that both chosen anthropogenic inventories additionally provide

sector-specific information. To include this information in our simulations, different sectors are linked to separate tracers. The





| This study | CAMS-GLOB-ANT (for $CO_2$ and $CH_4$) | REAS (for CO) |
|---|---|---|
| Energy | Power generation (ene) | Power plants point |
| | Fugitives (fef) | Power plants non-point |
| | Oil refineries and transformation sector (ref) | |
| Industry | Industrial processes (ind) | Industry |
| Transport | Road transportation (tro) | Road transport |
| | Off Road transportation (tnr) | Other transport |
| | Ships (shp) | |
| Residential & Waste | Residential, commercial and other combustion (res) | Domestic |
| | Solid waste and waste water (swd) | |
| Agriculture | Agriculture soils (ags) | |
| | Agricultural waste burning (awb) | |
| | Agriculture livestock (agl) | |

**Table 2.** Overview of mapping between the five broad sectors used in this study (first column) and the emission sectors provided by CAMS-GLOB-ANT v5.3 (second column) and REAS v3.2.1 (third column).

11 sectors from CAMS-GLOB-ANT were aggregated into five broad sectors to make the model simulations computationally less expensive. A similar aggregation was performed on the REAS sectors. The mapping is given in Table 2. This will allow us to track the respective contributions to the total simulated concentrations of the following source categories: energy, industry, transportation, residential & waste and agriculture. More detail about what is included in every sub-sector can be found in the
documentation of the respective data set.

Further, biomass burning emissions are coming from the Fire INventory from NCAR (FINN v2.5, Wiedinmyer et al. (2011)) for all species. The observation-based global $pCO_2$ climatology from Landschützer et al. (2017) is used to represent the ocean-atmosphere exchange of $CO_2$, while the $CH_4$ fuxes from wetlands are taken from the WetCHARTS v1.0 climatology (Bloom et al., 2017). Finally, WRF-GHG calculates the biogenic $CO_2$ fluxes online based on the Vegetation Photosynthesis and
Respiration Model (VPRM, Mahadevan et al. (2008); Ahmadov et al. (2007)). It uses its own calculated 2 m temperature and downward shortwave radiation together with surface reflectance data from the Moderate Resolution Imaging Spectroradiameter (MODIS) onboard the Aqua and Terra satellites. The extra required parameters for VPRM are taken from Li et al. (2020).

## 2.4    Comparing observations with WRF-GHG simulations

### 2.4.1    Xianghe in situ observations

The WRF-GHG model cell which covers the location of the instrument is selected to compare with the in situ observations. Because the concentrations are measured at an altitude of 60 m.a.g.l., this WRF-GHG profile is interpolated to that altitude,



using the model surface as ground level. Finally, the observations are averaged over a period of 30 minutes around the hourly model output.

### 2.4.2 Xianghe TCCON remote sensing observations

The same model cell as for the in situ observations is used to compare with the column observations. The five TCCON observations that are closest in time with the WRF-GHG output, but deviate no more than 15 minutes, are averaged and used for the comparison. The model profile is extended above 50 hPa with the TCCON a priori profile and then smoothed by using the averaging kernels in order to account for the instrument and retrieval characteristics (Rodgers and Connor, 2003). Note that an alternative approach would be to extend the model profiles with the CAMS reanalysis that is used as initial and lateral

boundary conditions. However, the accuracy issues with CAMS $CH_4$ data in the stratosphere are well-documented (Ramonet et al., 2021; Agustí-Panareda et al., 2023), and would introduce known biases into our study. Moreover the optimized a priori profiles of the TCCON GGG2020 data show improved accuracy in the stratosphere (Laughner et al., 2023), supporting our decision to utilize this data for extending the model profiles.

### 2.4.3 TROPOMI observations

To compare the spatial $XCH_4$ distribution of TROPOMI with those of WRF-GHG, the model profiles are extended above 50 hPa with the TROPOMI a priori column number density profiles of $CH_4$ and dry air (mol m$^{-2}$) to ensure that both products in the comparison cover the same altitude range. Since a typical $CH_4$ profile shows a sharp decrease in the upper layers of the atmosphere, this part has a non-negligible impact on the column-averaged mole fraction. Further, the extended WRF-GHG $CH_4$ profiles are smoothed with the TROPOMI column averaging kernels and a priori profiles following Apituley et al. (2023).

The column number density profiles of $CH_4$ and dry air are calculated from the hourly 3-D WRF-GHG output as follows:

$$\rho_i^{CH_4} = \nu_i^{CH_4} \rho_i^{da}, \text{ with } \rho_i^{da} = \frac{P_i}{RT_i} \frac{1}{1 + 1.6075 q_i} \tau_i. \tag{1}$$

In the above equation $\nu_i^{CH_4}$ is the $CH_4$ dry air volume mixing ratio (ppb) and $\rho_i^{da}$ the dry air column number density in WRF-GHG layer $i$. The dry air column number density $\rho_i^{da}$ is calculated according to the ideal gas law, where $P_i$, $T_i$ and $q_i$ are the air pressure (Pa), temperature (K) and water vapour mixing ratio with respect to dry air (kg kg$^{-1}$), respectively. The thickness

of layer $i$ (m) is represented by $\tau_i$. Finally, $R$ is the ideal gas constant 8.3145 J K$^{-1}$ mol$^{-1}$. Note that 1.6075 is the ratio of the molar mass of dry air with respect to the molar mass of water to convert wet air to dry air.

TROPOMI has an equator crossing time of around 13:30 local solar time, so we compute the equivalent simulated $XCH_4$ by taking the average over 12h-15h LT. Note that we use the model simulations from the d02 domain (which has a horizontal resolution of $9\times9$ km$^2$) for this analysis, instead of d03 as for the comparisons with Xianghe observations, since a larger spatial

extent is advantageous for a statistically effective comparison with TROPOMI.

Using the HARP toolset (part of the Atmospheric Toolbox, https://atmospherictoolbox.org/) for TROPOMI and the CDO software (Schulzweida, 2020) for WRF-GHG, both $XCH_4$ products are then binned to a common spatial grid to enable a quantitative analysis: we have chosen a regular latitude-longitude grid with a horizontal resolution of 0.05°.



# 3    Results and discussion

## 3.1    Overall model performance

With the model settings as elaborated in Sect. 2.3, WRF-GHG was run from 15 August 2018 to 1 September 2019. However, the first two weeks were regarded as a spin-up phase, so the analysis is made on one full year of data: from 1 September 2018 until 1 September 2019. This conservative spin-up period is implemented to ensure thorough mixing of the tracers within the domain. The complete data set can be accessed on https://doi.org/10.18758/P34WJEW2 (Callewaert, 2023).

An overview of the simulated and observed time series of the $CH_4$ concentrations at Xianghe is shown in Fig. 2, together with the model error. For the column observations, the model shows a mean underestimation of -3.03 ppb, with a moderate correlation of 0.56 (Table 3). At the surface level, the data were divided into afternoon (13:00–18:00 LT) and nighttime (03:00–08:00 LT) periods for statistical analysis, as models generally perform better in simulating concentrations during the afternoon when the lower atmosphere is better mixed. The definition of these time periods is based on the daily maximum and minimum values, as will be discussed later in Sect. 3.4.2. Indeed, the correlation is higher during the afternoon (0.66) compared to nighttime (0.42), with mean bias errors of 14.22 ppb and 12.68 ppb, respectively (see Table 3). Additionally, significantly larger errors were found at night, indicating greater challenges for the model in accurately capturing nighttime values compared to afternoon.

The moderate correlation coefficients are likely due to a seasonality in the bias: WRF-GHG is underestimating the $CH_4$ data at Xianghe in summer and autumn (June - November) and slightly overestimating them in winter (January - March), which is especially visible for the column data in Fig. 2c.

|      | insitu $CH_4$ (afternoon) | insitu $CH_4$ (night) | $XCH_4$ |
|------|---------------------------|------------------------|---------|
| BIAS | 14.22                     | 12.68                  | -3.03   |
| RMSE | 159.74                    | 334.06                 | 23.96   |
| CORR | 0.66                      | 0.42                   | 0.56    |

**Table 3.** Statistics of the model-data comparison of the ground-based $CH_4$ observations at the Xianghe site from 1 September 2018 until 1 September 2019. We present the mean bias error (BIAS), root mean square error (RMSE) and Pearson correlation coefficient (CORR). The mean bias error and root mean square error are given in ppb. For in situ observations, the data is split in afternoon (13-18 LT) and night (3-8 LT) hours.

Possible sources of this bias are inaccuracies in the background values, misrepresentation of $CH_4$ sources and sinks within WRF-GHG, or a combination of these factors. Given $CH_4$'s long atmospheric lifetime, background values significantly contribute to the total simulated signal, as also illustrated by the mean values for the background and total simulated tracer in the top of Table 4. We therefore start by further examining the global CAMS reanalysis which is used to represent the inflow and outflow at the model domain boundaries. In the CAMS validation report by Ramonet et al. (2021), a similar seasonal bias between CAMS $CH_4$ and TCCON is found. To explore this pattern in more detail and include Xianghe in the analysis, we



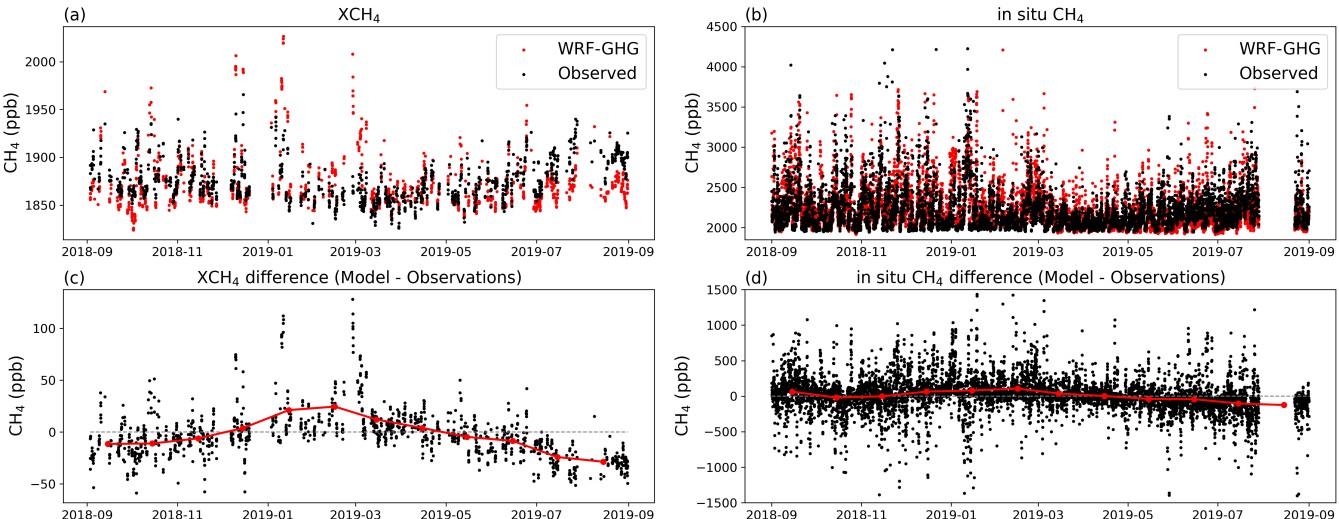

**Figure 2.** Time series of the observed (black) and simulated (red) (a) XCH$_4$ and (b) insitu CH$_4$ concentrations at the Xianghe site. Panels (c) and (d) show the differences between WRF-GHG simulations and observations for XCH$_4$ and in situ CH$_4$, respectively. Data points are hourly. The red points in (c) and (d) represent the monthly mean differences.

| | XCH$_4$ (ppb) | | | | in situ CH$_4$ (ppb) | | | |
| --- | --- | --- | --- | --- | --- | --- | --- | --- |
| | Q1 | median | mean | Q3 | Q1 | median | mean | Q3 |
| Total | 1900.74 | 1916.18 | 1927.36 | 1942.42 | 2028.75 | 2132.50 | 2212.50 | 2302.58 |
| Background | 1885.79 | 1890.75 | 1891.34 | 1896.57 | 1912.83 | 1927.69 | 1925.37 | 1938.20 |
| Biomass burning | 0.00 | 0.00 | 0.00 | 0.00 | 0.00 | 0.00 | 0.00 | 0.00 |
| Energy | 2.61 | 11.13 | 19.04 | 28.90 | 12.03 | 49.67 | 105.88 | 135.21 |
| Residential (& waste) | 2.65 | 5.86 | 8.17 | 10.72 | 31.24 | 65.49 | 94.33 | 122.11 |
| Industry | 0.07 | 0.17 | 0.21 | 0.30 | 0.77 | 1.63 | 2.29 | 2.99 |
| Transportation | 0.06 | 0.12 | 0.15 | 0.20 | 0.66 | 1.40 | 2.00 | 2.58 |
| Agriculture | 2.00 | 4.75 | 7.56 | 9.49 | 24.95 | 51.77 | 76.08 | 97.30 |
| Wetlands | 0.02 | 0.12 | 0.56 | 0.62 | 0.09 | 0.66 | 4.17 | 3.91 |
| Termites | 0.17 | 0.29 | 0.34 | 0.46 | 1.13 | 2.02 | 2.37 | 3.17 |
| Total tracers | 9.86 | 24.76 | 36.02 | 51.21 | 102.75 | 209.15 | 287.13 | 377.48 |

**Table 4.** Statistics of the total simulated CH$_4$ concentrations and the different tracer contributions over the complete simulation period. Q1 and Q3 represent the first and third quartile, respectively, between which 50 % of the data fall.

reproduce their calculations for several TCCON sites at similar latitudes (Karlsruhe (49.1° N), Orleans (48.0° N), Garmisch
(47.5° N), Park Falls (45.9° N), Rikubetsu (43.5° N), Lamont (36.6° N), Tsukuba (36.0° N), Edwards (35.0° N), Pasadena (34.1° N), Saga (33.2° N), and Hefei (31.9° N)) for the period of interest, as shown in Fig. 3. Indeed, we find a seasonal



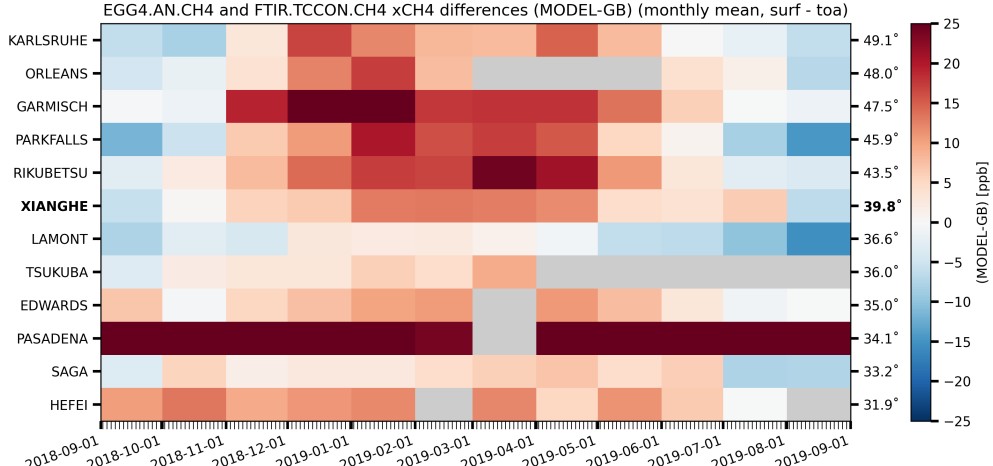

**Figure 3.** Monthly mean difference (in ppb) between CAMS reanalysis model and TCCON XCH$_4$ between 30 - 50° N over the simulation period of this study.

bias where CAMS is overestimating TCCON XCH$_4$ from December until May and showing a small underestimation in the rest of the period. The bias at Xianghe ranges from 13.17 ppb in February 2019 to -6.56 ppb in August 2019 (monthly mean differences). The monthly mean bias of WRF on the other hand, ranges between 24.49 ppb in February 2019 and -28.70 ppb in

August 2019 and shows a significantly larger amplitude than the CAMS bias. Moreover, the same seasonal pattern is found in the time series of the differences for the in situ data (Fig. 2d). Ramonet et al. (2021) assume the seasonal bias within CAMS is related to an inaccurate representation of the seasonal cycle of surface emissions and/or the OH sink. Similarly, the remaining WRF-GHG bias likely arises from errors in the seasonality of the CH$_4$ emissions and/or neglecting the reaction of CH$_4$ with OH. This will be further investigated in Sect. 3.3.

In the rest of this work, we have applied a bias correction to the WRF-GHG simulations by subtracting the monthly mean difference between CAMS and TCCON XCH$_4$, averaged over all sites (except Pasadena due to outlier behavior) between 30 - 50° N, from the background tracer. The updated statistical metrics are given in Table 5. The correlation coefficient for the column data slightly improves to 0.65, where for the surface concentrations the bias correction has only a negligible impact on the model-data comparison. The remaining monthly mean bias for XCH$_4$ (in situ CH$_4$) ranges between 13.04 ppb (95.20 ppb)

in February 2019 and -25.70 ppb (-121.25 ppb) in August 2019, which is still larger than the measurement uncertainty of 6 ppb (1 ppb).

## 3.2   Sector contributions to observed concentrations

All fluxes that are included in WRF-GHG are tracked in separate tracers, as explained in Sect. 2.3. This allows us to disentangle the total simulated concentrations into the different tracer contributions and evaluate the influence of different source sectors

on the observations at Xianghe, as well as their respective importance. An overview of the monthly mean values is shown in




|  | insitu CH$_4$ (afternoon) | insitu CH$_4$ (night) | XCH$_4$ |
|---|---|---|---|
| BIAS | 8.43 | 6.88 | -8.10 |
| RMSE | 158.29 | 333.22 | 22.35 |
| CORR | 0.66 | 0.43 | 0.67 |

**Table 5.** Same as Table 3 but with bias corrected model values.

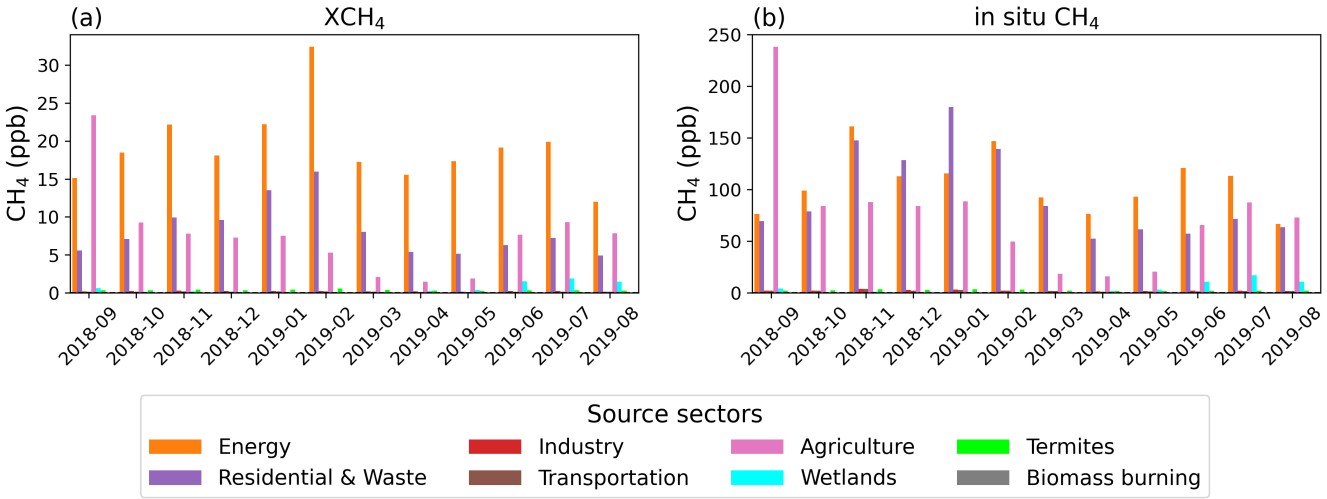

**Figure 4.** Monthly mean tracer contributions above the background for (a) XCH$_4$ and (b) in situ CH$_4$ simulated concentrations at Xianghe.

Fig. 4, while additionally the median and interquartile range of the complete period are given in Table 4. Note that all simulated hours were used for this analysis, not just the ones coinciding with observations.

For CH$_4$, the simulated signal at Xianghe is mainly determined by three sectors: energy, residential & waste (which com-
bines both residential heating and waste management sectors) and agriculture. They respectively contribute with a median enhancement of 11.13 ppb, 5.86 ppb and 4.75 ppb above the background for the columns and 49.67 ppb, 65.49 ppb and 51.77 ppb near the surface (see Table 4). Furthermore there is a small contribution from wetlands in summer, peaking in July with a median tracer contribution of 1.49 ppb for the columns and 10.65 ppb near the surface. Other sectors such as industry, transportation, termites and biomass burning seem to be irrelevant at Xianghe. Overall, the total tracer enhancement is about ten
times larger for the in situ concentrations compared to the column-averaged values.

The fact that the dominant source sectors (agriculture, residential heating, waste management and energy (which is mainly coal mining in this case)) are not known for releasing CH$_4$ at elevated altitudes, supports our choice to implement the emissions only in the lowest model layer.

Futhermore, remark that for the in situ concentrations, the three dominant sectors are roughly equally important, while for



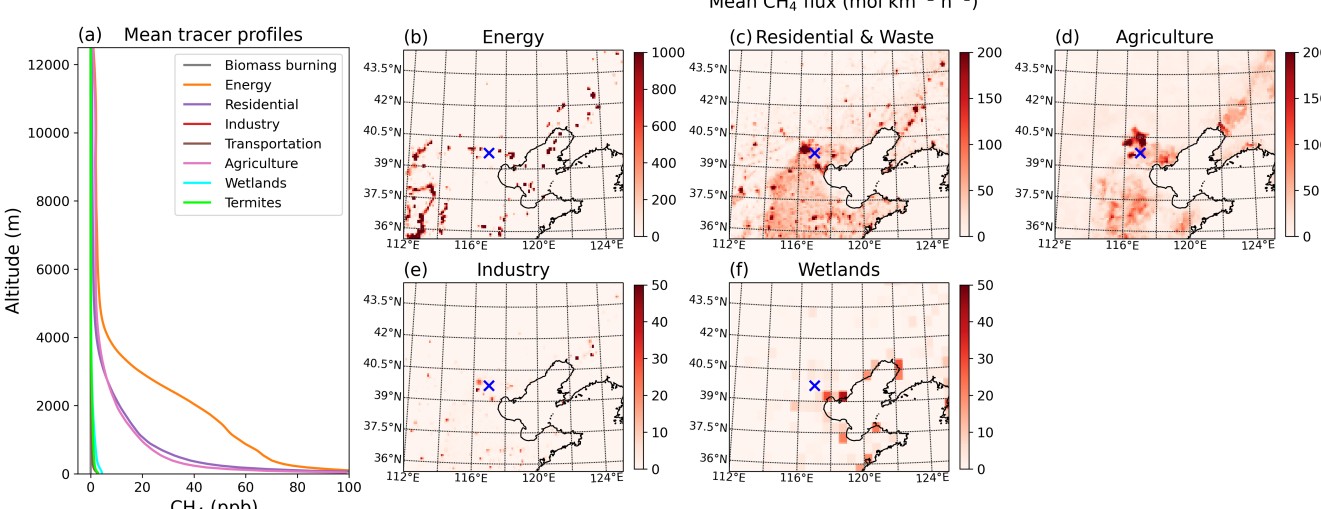

**Figure 5.** (a) Mean vertical profile of the tracer fields in WRF-GHG for $CH_4$ at Xianghe. All simulated hours were used for this plot. (b-f) Maps of the mean $CH_4$ flux (mol $km^{-2}$ $h^{-1}$) in WRF-GHG domain d02 during the entire simulation period for the most important sectors. Remark that different sectors have different ranges in the colorbar. The lcoation of the Xianghe site is indicated by the blue cross.

the column concentrations we find a larger impact of the energy sources: the relative mean enhancement of the energy tracer is 52.87% for the column concentrations, while it is only 36.88% for the surface concentrations. When looking at the mean vertical profiles of the different tracer contributions above Xianghe (Fig. 5a) we see that the contributions from the energy sector are generally found at a higher altitude compared to other sectors. High concentrations near the surface are associated with emission sources nearby, while those aloft are likely caused by long-distance pollutant transport in the free troposphere. Therefore, we assume that this difference between column and surface energy contribution is because the strongest energy sources are situated in Shanxi (the largest coal producing province in China), which is much further away from Xianghe than for example the strongest residential (mainly Beijing and Tianjin) and agricultural sources, see Fig. 5b-d.

In Fig. 4, we further observe a larger residential signal in winter, where the median tracer contribution peaks with 13.43 ppb in February for the columns and with 132.75 ppb in January, near the surface. Meanwhile, the influence from agriculture reaches its maximum in September (monthly median values of 14.46 ppb for $XCH_4$ and 196.07 ppb for in situ $CH_4$) and its minimum in March-April (monthly median values of 0.89 ppb for $XCH_4$ and 11.49 ppb for in situ $CH_4$). This corresponds with the seasonal pattern of emissions within CAMS-GLOB-ANT.

## 3.3 Seasonal $CH_4$ bias

In Sect. 3.1, we identified a seasonal bias in the $CH_4$ simulations (WRF-GHG underestimates $CH_4$ in summer and autumn, overestimates in winter) that could not be fully explained by a similar bias in the background data, indicating a potential bias in the seasonality of the emission data and/or a consequence of ignoring the OH sink. In this section, we first investigate the





primary emission sectors that may have contributed to this seasonal bias. One of the major sources of $CH_4$ at Xianghe is the energy sector (see Sect. 3.2), primarily through fugitive emissions from the extraction, processing, storage, and transport of coal, oil, and natural gas. These emissions are not expected to exhibit significant monthly variation. Indeed, the energy emissions in the CAMS-GLOB-ANT inventory are relatively stable throughout the year: they show a coefficient of variation (CV, calculated as the ratio of the standard deviation to the mean) of only 0.42% for the monthly averaged values across the model domain. As a result, our focus will be on the following emission categories: agriculture, residential & waste, and wetlands.

- Agriculture. As presented in Table 2, the agricultural sector is comprised of three subsectors: soils (this is mainly rice cultivation), agricultural waste burning, and livestock (manure management and enteric fermentation). In China, rice cultivation plays a vital role but is predominantly concentrated in regions south of 35°N. In CAMS-GLOB-ANT, the most important agriculture subsector in the region of the Xianghe site is livestock. According to the emission inventory, livestock emissions in the wide region around Xianghe peak in September and reach their lowest levels in March and April. Unfortunately, the source of these monthly variations in $CH_4$ emissions within the inventory is unclear, as the accompanying data set of temporal factors, CAMS-GLOB-TEMPO (Guevara et al., 2021), references constant factors for $CH_4$ emissions from agricultural sources. Previous research by Maasakkers et al. (2016) suggests that emissions from manure management often correlate with air temperature, with higher emissions during warmer months (May to September in this case) and lower emissions during colder months (December to February). If the true seasonality of agricultural emissions around Xianghe is indeed temperature-driven, this implies that the current inventory underestimates emissions during spring and summer (May to August) and overestimates them in winter, as it shows a peak only in September and a minimum in spring (March-April) rather than in winter. This discrepancy in the seasonality of emissions could explain the seasonal bias observed in our $CH_4$ simulations, pointing to inaccuracies in the representation of agricultural emissions.

- Residential & waste. This sector represents emissions from residential, commercial and other combustion sources together with $CH_4$ emissions from solid waste and waste water treatment. In CAMS-GLOB-ANT, the waste sector is the most important one in the Xianghe region and assumed to be relatively constant throughout the year: monthly total $CH_4$ emissions between 38-41 °N and 115-119 °E range between 0.0408 Tg and 0.0452 Tg. In summer, total residential combustion emissions in the region can be as low as 0.0039 Tg per month, while in winter, they are almost of the same size as the waste emissions: 0.0357 Tg. So the seasonality of the residential & waste sector is coming from the residential part, peaking in winter. However, Hu et al. (2023) showed that $CH_4$ emissions from waste treatment often follow the seasonality of air temperature. Even though this study is based on observations in the Hangzhou megacity, their results could possibly be representative for the BTH region as well. This would mean that the waste emissions are underestimated in summer and/or overestimated in winter, which would match the current model-observation mismatch for $CH_4$.

- Wetlands. Within the WRF-GHG simulations, wetlands only show minor contributions to the surface and column data, and only in summer. Emissions are taken from the WetCHARTs v1.0 ensemble data set. In the BTH area, the main





wetland areas are located close to the Bohai Sea (see Fig. 5f). However, according to WetCHARTs, these emissions are relatively small compared to those from wetlands more in the south of China. In an evaluation of the WetCHARTs ensemble against GOSAT observations by Parker et al. (2020), a general underestimation of the seasonal amplitude in China was found. Furthermore, Chen et al. (2022) showed increased posterior wetlands emissions compared to the a priori values when inferring yearly CH$_4$ emissions over China using TROPOMI satellite observations. This could point to an underestimation of the wetland emissions in the current study, and therefore an underestimation of CH$_4$ in summer.

The observed seasonal error pattern between the WRF-GHG CH$_4$ simulations and the Xianghe observations may be due to one or more of the reasons previously mentioned. To gain a spatial perspective on this seasonal bias, we compared the WRF-GHG XCH$_4$ field with TROPOMI observations. Figure 6 shows the seasonal mean XCH$_4$ from both WRF-GHG and TROPOMI, as well as their normalized difference over the broader Xianghe region. To highlight seasonal variations, we subtracted the mean difference between WRF-GHG and TROPOMI over the entire simulation period (also shown in Fig. 12d) from the seasonal means, resulting in a 'normalized difference.' Overall, we find a mean bias error between WRF-GHG and TROPOMI of -10.55 ppb (or -0.56% [(TROPOMI - WRF-GHG)/WRF-GHG]), consistent with previous studies. (Yang et al., 2020; Tian et al., 2022; Sha et al., 2021).

The analysis reveals a model underestimation in summer (JJA) and an overestimation in winter (DJF), see Fig. 6. The biases are smaller in spring and autumn. However, we cannot identify a distinct spatial pattern throughout the seasons that could point to errors within a specific source sector. Figure 6 shows differences on a large spatial scale, suggesting that for example the underestimation by WRF-GHG is linked to emission sources that are widespread in the region. Since the North China Plain is a livestock-dominated region with strong urbanization and industrial activities, this implies that the fluxes of either agriculture (livestock), waste treatment, or both, rather than the fluxes from wetlands, are underestimated in summer in CAMS-GLOB-ANT. Given the lack of a clear outcome from our analysis, it is likely a combination of factors.

Finally, we used backward simulations with the FLEXible PARTicle dispersion model (FLEXPART) v10.4 (Pisso et al., 2019) to evaluate the impact of the OH sink on CH$_4$ concentrations at Xianghe. Details of the model configuration are provided in Appendix B. By comparing simulations that include or exclude the chemical reaction with OH, we estimated its influence. The results indicate a more pronounced difference in summer than in winter, with mean relative backward sensitivity differences of about 0.04%, 0.005%, 0.05%, and 0.2% in October 2018, January 2019, April 2019, and July 2019, respectively, over the entire footprint. Considering the size of CH$_4$ emissions within the WRF-GHG domain (Table 4), the contribution of the ignored OH reaction to the CH$_4$ mole fraction is around 0.11 ppb in winter and 4.4 ppb in summer, which remains small compared to the measurement uncertainties (1 ppb for in situ data and 6 ppb for TCCON) and the magnitude of the observed bias. Moreover, the higher impact in summer should theoretically cause a model overestimation during this season if the OH sink is ignored. However, since the observed seasonal bias shows a different trend, it is unlikely to be driven by CH$_4$ chemistry.

Therefore, our analysis highlights an urgent need for further research into the seasonality of CH$_4$ emissions in northern China.







**Figure 6.** Seasonal mean XCH$_4$ (ppb) over the domain d02 (provinces of Beijing, Tianjin, Hebei, Shanxi and part of Shandong) as simulated by WRF-GHG (first column) and observed by TROPOMI (second column), as well as the normalized difference between them (WRF-GHG - TROPOMI, in ppb). Normalized difference indicates that the mean difference over the entire simulation period is subtracted from the seasonal means. The seasons are defined as (a,b,c) SON: September - November (autumn), (d,e,f) DJF: December - February (winter), (g,h,i) MAM: March - April (spring) and (j,k,l) JJA: June - August (summer). White pixels indicate that there are no observations available during the entire period.




### 3.4 Impact of meteorology on variability of concentrations

In Sect. 3.2, we showed how emissions from different sources affect the $CH_4$ observations at Xianghe. In the current section we want to focus on the meteorological factors that influence the temporal variability of the time series. More specifically we will discuss the impact of large-scale phenomena, the planetary boundary layer and local winds.

#### 3.4.1 Synoptic scale winds

Because FTIR observations generally have a large area of representativeness (generally a few 100 km), column concentrations are relatively insensitive to local fluxes and vertical mixing, while they are strongly influenced by large-scale patterns (Keppel-Aleks et al., 2011). We use the winds at 800 hPa to represent horizontal transport in the free troposphere, as this altitude is generally above the planetary boundary layer height. More specifically, we looked at the daily mean column concentrations above the background for every wind direction to see if a clear relationship could be found. This is shown in Fig. 7. Remark that only southwest (SW) and northwest (NW) wind segments are given because southeast and northeast winds occur only seldom at 800 hPa: only on 2 and 13 days out of 231, respectively. We find that in general, larger enhancements are found when winds blow from the SW wind segment (median tracer contribution of 57.79 ppb) compared to the NW segment (median tracer contribution of 7.33 ppb). To quantify the difference, we conducted a non-parametric Mann-Whitney U test on the two categories, which yielded p-values well below 0.05 (see Fig. 7, in the title), indicating that the differences are statistically significant. Higher concentrations coincide with 800 hPa winds coming from the SW while NW winds correspond with lower concentrations. Yang et al. (2020) already showed that the day-to-day variation of the column observations of $CH_4$, $CO_2$ and CO are highly intercorrelated, and that clean days are linked with air from the north, while polluted days are linked with air from the south, which is confirmed here by the WRF-GHG simulations. Air masses from the north have been moving over rather remote and clean areas such as Inner Mongolia, Mongolia and Russia. Meanwhile, southerly air is linked with the highly populated North China Plain (NCP), where many anthropogenic emission sources are located.

The influence of polluted air from the southwest is visible in the surface concentrations as well, as we find a high correlation coefficient of 0.79 between the daily mean column and surface tracer enhancements. This indicates that both surface and column $CH_4$ concentrations are affected by synoptic-scale winds, which advect either clean or polluted air masses to Xianghe. Furthermore, the levels of pollution in these air masses can vary significantly from month to month due to changing meteorological conditions. For instance, during the winter months, weather conditions are generally more favorable to the accumulation of pollutants, leading to higher pollution levels (Li et al., 2022). This can intensify both local pollution plumes and those transported by southwestern winds. This phenomenon likely explains why despite relatively constant emissions throughout the year, we observe a significant month-to-month variability in the energy tracer contributions (see Fig. 4). More specifically, we find a CV of 26.53% for the column and 26.65% for the surface tracers. These findings suggest that tracer concentrations at Xianghe result from a complex interplay of emissions, wind direction, and weather patterns both near and far.





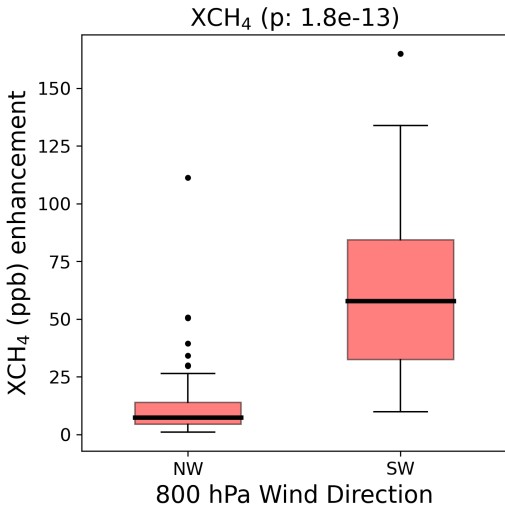

**Figure 7.** The distribution of the daily mean simulated column tracers above the background per 800 hPa wind direction category and species. NW is for winds with an angle of 292.5 to 337.5 ° from north, while SW represents the angles between 202.5 and 247.5°. There are 72 days with NW winds and 33 days with SW winds. The colored boxes indicate the range between the first and third quartile, while the thick solid line is the median. Outliers (values that are 1.5 times the interquartile range above (below) the third (first) quartile) are shown by black dots.

### 3.4.2 Planetary boundary layer dynamics

The planetary boundary layer (PBL) is the lowermost layer of the atmosphere which is in direct contact with the Earth's surface. The characteristics of this layer vary throughout the day. During the day, under influence of solar radiation, turbulent motions cause strong vertical mixing of the air within the PBL. These processes allow gases to be dispersed and transported upwards, which generally leads to reduced concentrations near the surface. At night, radiational cooling of the surface creates a temperature inversion close to the ground. This causes the nocturnal PBL to be stable and more shallow, trapping pollutants near the surface and as such increasing their local concentrations.

Figure 8 shows the diurnal variation of the PBL height as simulated by WRF-GHG and the $CH_4$ concentrations near the surface (both simulated and observed). Indeed, the height of the PBL in WRF-GHG is largest in the afternoon when solar radiation is strongest, reaching its peak at 15:00 (local time). This corresponds with the lowest simulated surface concentrations (Fig. 8b), where we find median (and interquartile) values of 2039.77 (1977.74 - 2158.28) ppb. Right after sunset, the height of the PBL drops to its lowest value ($\approx$ 50 m - 430 m in WRF-GHG), after which it persists during the course of the night, until sunrise. This period corresponds with slightly increasing $CH_4$ concentrations as emissions near the surface accumulate within this stable shallow layer. Hence, the highest concentrations are found in the early morning: at 8:00 with 2239.75 (2079.29 - 2484.04) ppb. As the PBL height starts to rise at 8:00 due to turbulent mixing, $CH_4$ start to drop in WRF-GHG, creating a diurnal cycle.

Note that WRF-GHG is quite capable at simulating this diurnal variation of $CH_4$ in situ observations. The observations show



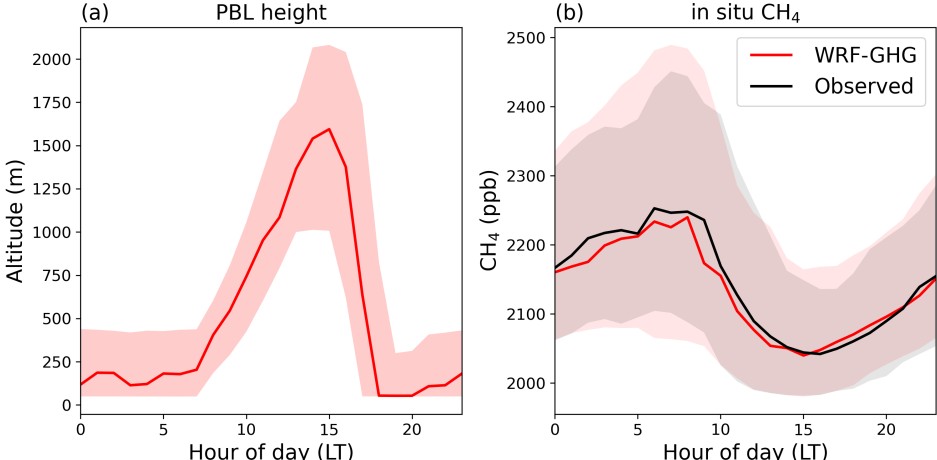

**Figure 8.** Hourly median and interquartile range of the (a) simulated planetary boundary layer height, and (b) observed and simulated surface CH$_4$ concentration at Xianghe.

minimal concentrations at 16:00 with a median (and interquartile) value of 2041.94 (1981.54 – 2135.88) ppb, which are well captured by WRF-GHG, even though one hour earlier. The peak CH$_4$ concentrations however, are observed at 6:00 with a median (and interquartile) value of 2252.71 (2104.36 – 2451.01) ppb, portraying a small model underestimation of about 13

ppb. Together, this leads to a small underestimation of the CH$_4$ diurnal amplitude in WRF-GHG of 10.79 ppb.

We have shown that these PBL dynamics are very important for the variability of the surface concentrations, however they are irrelevant for the column concentrations, as the latter are much less affected by vertical transport (Wunch et al., 2011). Indeed, the WRF-GHG simulated column concentrations don't exhibit a clear diurnal cycle, suggesting that this aspect is well captured by the model. It is however difficult to validate this using observations, as FTIR measurements are only possible during periods

of sunlight.

### 3.4.3 Local emissions

Regional emissions are influencing both column and in situ concentrations at Xianghe, as elaborated in Sect. 3.4.1. However, emission sources nearby could also have an impact on these values, especially for the in situ observations as they sample the local air. To analyze which nearby sources influence the Xianghe measurements, we look for correlations between the 10m

wind speed and direction and the simulated concentrations. Figure 9 reveals the mean WRF-GHG tracer contribution per wind direction and speed for CH$_4$. To eliminate the influence of polluted plumes from further away, we select only those days on which the mean daily XCO enhancement is smaller than 45 ppb. We use XCO as a tracer for polluted events as it is the species with the shortest atmospheric lifetime. Furthermore, we compute the mean concentrations separately for day and night to avoid the effects of the PBL. The night hours are defined as those with the peak concentrations, i.e., between 3h and 8h LT, while

the day represents those hours with highest atmospheric mixing and lowest concentrations, i.e., between 13h and 18h. During



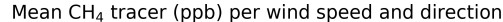

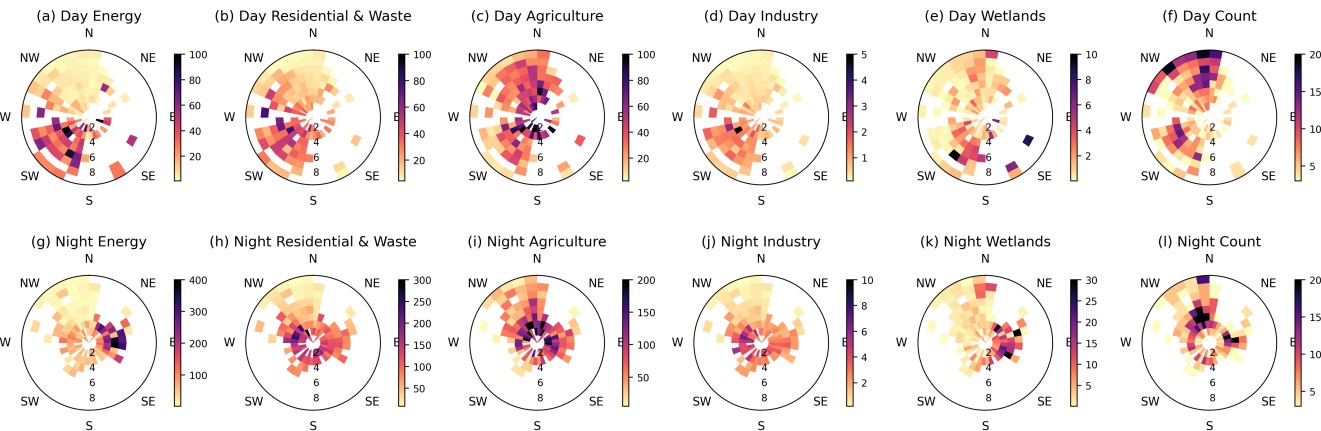

**Figure 9.** Mean CH$_4$ simulated tracer concentrations (indicated by colour scale, in ppb) binned per wind speed and direction for the main sectors (a) energy, (b) residential & waste, (c) agriculture, (d) industry and (f) wetlands on days without strong regional pollution. The first row represents afternoon hours (13h - 18h LT), while the second row represents nighttime hours (3h - 8h LT). Data is binned per 1 m s$^{-1}$ and 11.25° wind direction. (f) Count of data points in each bin. Only bins with at least 3 points are included in the figure. Remark that the panels have different colour scales.

the day most winds are coming from the north and southwest, while at night the most frequent wind directions near the surface are north and east. Higher wind speeds are found during the day than at night. The northern winds typically have the lowest tracer contributions since there are fewer emission sources in this direction, with the exception of agriculture (see Fig. 10). In general, we see that wind directions with the largest enhancements correspond with the largest sources nearby (Fig. 9-10):

east and west for energy, all but north for residential, all directions for agriculture, southwest for industry and southeast for wetlands. The highest values overall ($>$ 400 ppb) are found for the energy tracer at night and they are coming from the east, where some very large CH$_4$ point sources are located that correspond to coal mine emissions nearby the city of Tangshan (see Fig. 10a). However, when looking closer at the CH$_4$ time series (not shown) we see that WRF-GHG is often overestimating the Xianghe in situ CH$_4$ observations at times where the model shows a large energy contribution. This is also visible in Fig.

11. This makes us to believe that these coal mine emissions might be overestimated in CAMS-GLOB-ANT. In the next section we further investigate this hypothesis by comparing WRF-GHG concentration fields with TROPOMI observations.

### 3.5 Source assessment near Tangshan

By comparing the yearly TROPOMI XCH$_4$ with WRF-GHG XCH$_4$, we want to assess if the CH$_4$ emissions from coal mines around Tangshan are indeed overestimated in CAMS-GLOB-ANT or not. Figure 12 shows the maps of the mean XCH$_4$ during

the entire simulation period: September 2018 until September 2019. The yearly mean total CH$_4$ fluxes from CAMS-GLOB-ANT in the WRF-GHG d02 is also given, as well as the difference between WRF-GHG and TROPOMI. By taking the average



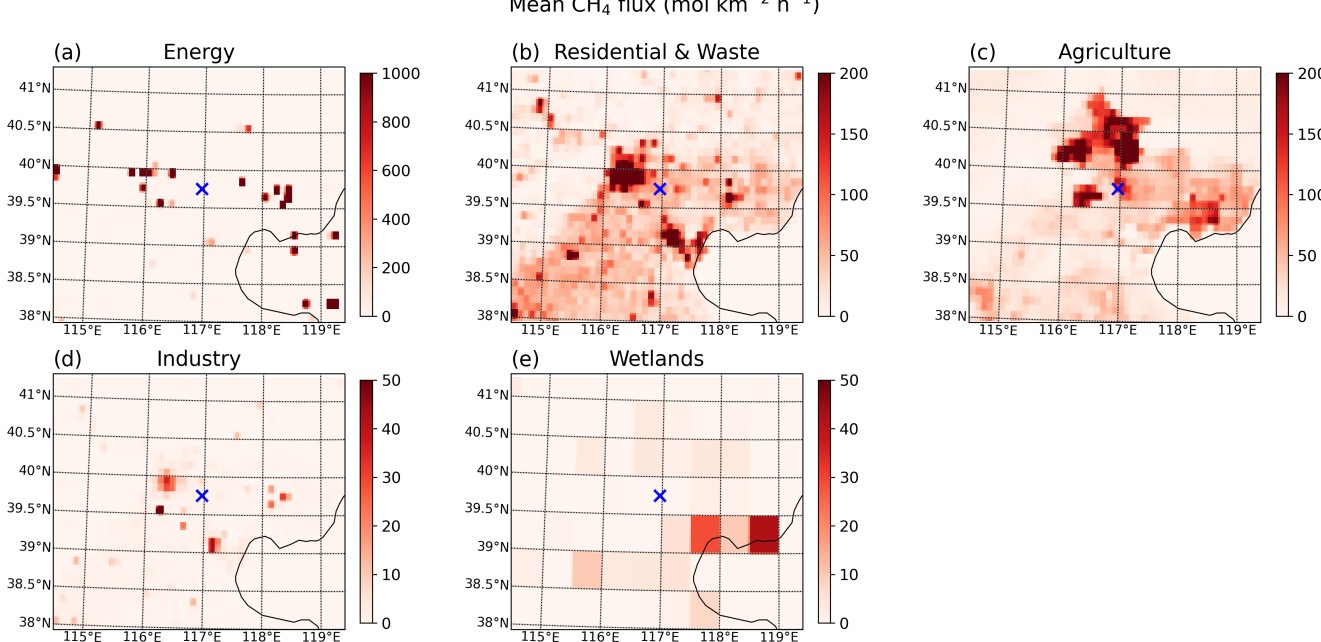

**Figure 10.** Map of the mean $CH_4$ flux (mol km$^{-2}$ h$^{-1}$) in WRF-GHG domain d03 during the entire simulation period from September 2018 until September 2019, for the most important sectors. Remark that the panels have different colour scales. The location of the Xianghe site is indicated by a blue cross.

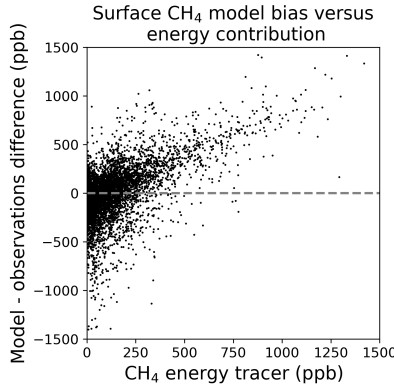

**Figure 11.** Correlation between energy tracer contribution to simulated $CH_4$ surface concentrations and differences between total simulated and observed surface concentrations. For this plot, the data was not filtered on day, night or polluted/clean days.



over the complete simulation period we minimize the influence of meteorological patterns on the XCH$_4$ concentration and expose the main emission sources.

When comparing the WRF-GHG input fluxes in Fig. 12a with the resulting XCH$_4$ concentration field in Fig. 12b, we indeed
find a strong agreement. The largest sources are found to the west of 114°E, which correspond to the extensive coal mining activities in Shanxi. In the same locations on the XCH$_4$ map of WRF-GHG we find the highest concentration values of the region. Unfortunately due to the mountainous terrain, TROPOMI observations are sparse in this area. Other sources, such as a hotspot around 36.25°N, 116.75°E and the slightly smaller emissions around Beijing (40°N, 116.3°E) and Tangshan (39.6°N, 118.4°E) correspond with elevated XCH$_4$ values. This suggests that yearly averaged XCH$_4$ maps can indeed reveal the strongest emis-
sion sources. It should be noted however, that the CH$_4$ sources around Beijing and Tangshan are approximately three times smaller than those in Shanxi (west of 114°E) and are barely strong enough to cause significant enhancements in the yearly XCH$_4$ maps. Our analysis indicates that point sources should emit at least around 0.1 Tg per year to be clearly distinguishable on annual XCH$_4$ maps, taken into account the noise of the observations. The region below 37°N shows high simulated XCH$_4$ values as well, however they do not directly correspond to strong sources in the inventory. This can likely be explained by the
presence of the Taihang mountains on the west which lead to poor dispersion conditions (Fu et al., 2014). Therefore the larger concentrations in this area are likely more determined by the topography and associated meteorological conditions than by surface fluxes.

We observe slightly elevated XCH$_4$ values near the coal mines of Tangshan in both the WRF-GHG (1888.39 ppb compared to 1883.12 ppb in the surrounding area) and TROPOMI (1879.01 ppb vs 1876.64 ppb) maps. The surrounding area is defined
between 39.3-40 °N and 117.8-118.8 °E as there are no major CH$_4$ sources located therein, while the coal mine sources are concentrated in the area between 39.45-39.8 °N and 118.15-118.6 °E. Although these differences are minor, the enhancement in WRF-GHG is somewhat greater than in TROPOMI. More specific, the mean difference between WRF-GHG and TROPOMI is 8.87 ppb around Tangshan, while it is 11.85 ppb near the emission sources, suggesting that the model overestimation is more pronounced over the coal mines of Tangshan compared to the surrounding area. This difference is statistically significant with a
p-value of 7e-10, according to a one-sample t-test. This analysis suggests that these emission sources are indeed overestimated in CAMS-GLOB-ANT, occasionally leading to an overestimation of the energy tracer at Xianghe.

Note that the XCH$_4$ maps in Fig. 12 suggest that it is very likely that the CH$_4$ hotspot around 36.25°N, 116.75°E is overestimated as well, as the very strong XCH$_4$ enhancement in WRF-GHG is absent in the TROPOMI map.

## 4 Conclusions

We have used the WRF-Chem model in its greenhouse gas option WRF-GHG to simulate surface concentrations and column abundances of CO$_2$, CH$_4$ and CO observed at the Xianghe site in China, aiming to improve our understanding of the variabilities in the measured time series. Since June 2018, column-averaged concentrations are measured with a FTIR spectrometer that is part of TCCON, while near-surface concentrations of CO$_2$ and CH$_4$ are measured with a PICARRO CRDS analyzer at an altitude of 60 m.a.g.l. We computed 3-D concentration fields from September 2018 until September 2019 in




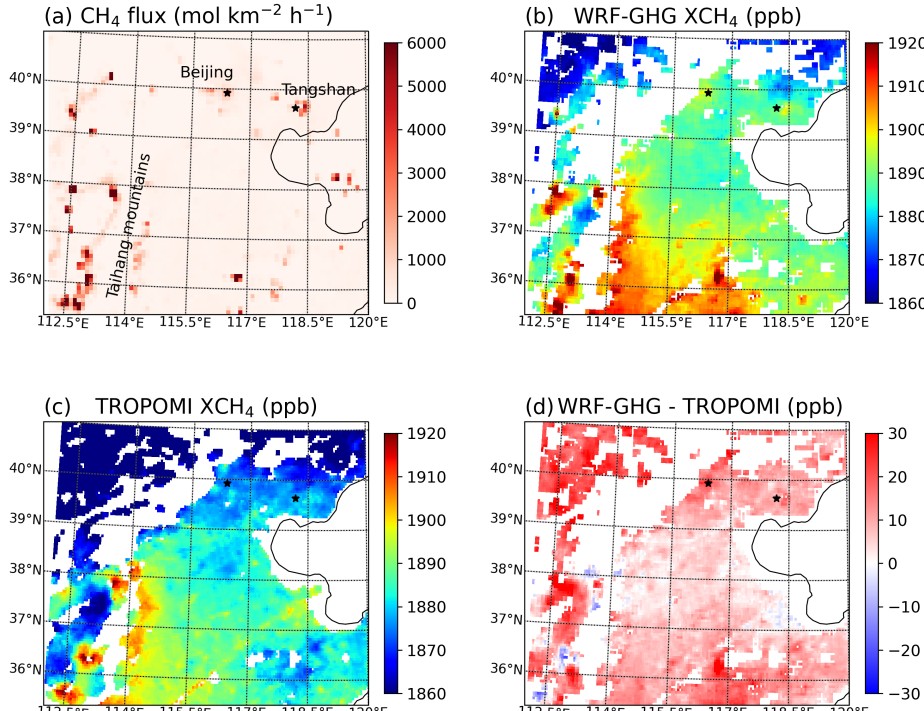

**Figure 12.** (a) The CH$_4$ flux from all sectors in CAMS-GLOB-ANT averaged from September 2018 until September 2019 and regridded to WRF-GHG grid d02 (9 km resolution). Mean XCH$_4$ over the same period as (b) simulated by WRF-GHG and (c) observed by TROPOMI (both regridded to 0.05 °). (d) Mean difference between WRF-GHG and TROPOMI XCH$_4$ over the entire simulation period.

three nested domains covering a large part of China. The ground-based observations are compared with simulations from the innermost domain, centered on the Beijing-Tianjin-Hebei region, with a horizontal resolution of 3×3 km$^2$. We employed the CAMS-GLOB-ANT v5.3 inventory for anthropogenic emissions of CO$_2$ and CH$_4$, and the REAS v3.2.1 dataset for CO. To disentangle the total simulated signal into the various source sectors, including a wide range of both natural and anthropogenic sources, they were simulated as separate tracers. This study is the first part of the analysis, focusing on CH$_4$.

In general, the model demonstrated moderate performance, with a correlation coefficient of 0.66 for near-surface CH$_4$ concentrations in the afternoon and 0.56 for column-averaged concentrations. After adjusting for the observed seasonal bias coming from the boundary conditions (CAMS reanalyses), the performance improved, as indicated by an increase in the correlation coefficient to 0.67 with the TCCON time series.

The simulated CH$_4$ concentrations is predominantly influenced by emissions from three main human activity sectors: energy, residential & waste, and agriculture. The energy sector has a more significant impact on column abundances (accounting for 52.9% of the total enhancement) compared to surface concentrations (36.9%), reflecting differences in the sensitivity of remote sensing and in situ measurements to sources at large distances, such as Shanxi province. For the in situ concentrations, the three emission sectors are equally important.





Monthly variability in the contributions from each tracer is found to align broadly with expected emission patterns: the residential tracer is higher in winter, while the agricultural tracer peaks in late summer (September). This month-to-month variation is further influenced by meteorological conditions such as horizontal advection and atmospheric stability, which is especially visible in the energy tracer where corresponding emissions remain relatively constant throughout the year, while the contributions from this tracer show notable variations.

The model simulations confirm the importance of large-scale wind patterns, with air masses from the southwest transporting higher $CH_4$ concentrations to the Xianghe site compared to those from the northwest (median tracer contributions of 57.8 ppb vs. 7.3 ppb, respectively). During southwest wind regimes, pollution from the densely populated North China Plain reaches the Xianghe site. While large-scale air masses influence the variability of both measurement types, smaller-scale factors such as planetary boundary layer dynamics and local wind patterns, also play a significant role for the near-surface concentrations. WRF-GHG effectively captures the diurnal variability driven by these boundary layer dynamics, with $CH_4$ surface concentrations reaching their lowest levels in the afternoon (16:00 LT) and peaking around sunrise (6:00 LT), leading to a diurnal amplitude of almost 200 ppb.

Despite correcting for the bias in boundary conditions, a residual seasonal bias remained in the model, likely due to inaccuracies in emission estimates from agricultural (livestock) and waste management activities. Furthermore, comparisons between simulated and observed $CH_4$ concentrations near the surface, along with TROPOMI $XCH_4$ data, indicate an overestimation of coal mine emissions near Tangshan in the emission inventory of CAMS-GLOB-ANT. However, due to the averaging effect in the column measurements and the relatively low emission strength, this source is just at the threshold of being distinguishable in the $XCH_4$ enhancements.

In summary, the WRF-GHG model successfully captures key aspects of $CH_4$ variability at the Xianghe site for both remote sensing and in situ observations. The model simulations also provide valuable insights into the relative contributions of different source sectors and the influence of meteorological processes on $CH_4$ concentrations.

However, the observed discrepancies, particularly the seasonal bias and overestimated emissions from certain sources, underscore the need for improved emission inventories in this region of China, especially for agricultural, waste management, and coal mining activities. Future research should aim to enhance our understanding of the monthly variations of $CH_4$ in northern China, which is crucial for providing more accurate boundary conditions and emission flux information to high-resolution modeling studies like the present work. By addressing these challenges, we can further refine our understanding of $CH_4$ sources and their impacts on regional air quality, ultimately contributing to more effective greenhouse gas mitigation strategies.

*Code and data availability.* The ERA5 and CAMS reanalysis data set (Hersbach et al., 2023a, b), used as input for the WRF-GHG simulations, was downloaded from the Copernicus Climate Change Service (C3S) Climate Data Store (2022). The CAMS-GLOB-ANT v5.3 emissions (Granier et al., 2019; Soulie et al., 2023) and temporal profiles CAMS-GLOB-TEMPO v3.1 (Guevara et al., 2021) are archived and distributed through the Emissions of atmospheric Compounds and Compilation of Ancillary Data (ECCAD) platform. The REAS emission inventory is publicly available at https://www.nies.go.jp/REAS/ (Kurokawa and Ohara, 2020). The WRF-Chem model code is distributed by



| Test | PBL | Surface Layer | Radiation |
|------|-----|---------------|-----------|
| BASE | YSU scheme (option 1) | Revised MM5 scheme (option 1) | RRTM and Dudhia (option 1) |
| A | YSU scheme (option 1) | Revised MM5 scheme (option 1) | RRTMG (option 4) |
| B | MYJ scheme (option 2) | Eta similarity scheme (option 2) | RRTMG (option 4) |
| C | MYNN3 scheme (option 6) | Eta similarity scheme (option 2) | RRTMG (option 4) |
| D | MYNN3 scheme (option 6) | Revised MM5 scheme (option 1) | RRTMG (option 4) |

**Table A1.** Overview of sensitivity tests on different physical parameterization options. They are a combination of three different PBL schemes: Yonsei University (Hong et al., 2006), Mellor-Yamada-Janjic (Janjić, 1994) and Mellor-Yamada-Nakanishi Niino Level 3 (Nakanishi and Niino, 2006, 2009; Olson et al., 2019); two surface layer schemes: Revised MM5 (Jiménez et al., 2012) and Eta similarity (Janjić, 1994); and two radiation schemes: RRTMG Longwave and Shortwave schemes (Iacono et al., 2008) versus RRTM Longwave and Dudhia Shortwave schemes (Dudhia, 1989; Mlawer et al., 1997).

NCAR (https://doi.org/10.5065/D6MK6B4K, NCAR, 2020). The WRF-GHG simulation output created in the context of this study can be accessed on https://doi.org/10.18758/P34WJEW2 (Callewaert, 2023). The TCCON data were obtained from the TCCON Data Archive hosted by CaltechDATA at https://tccondata.org (Zhou et al., 2022), while the surface observations at Xianghe were received through private communication with the co-authors. TROPOMI Level 2 Methane Total Column data are publicly available online at https://doi.org/10.5270/S5P-3lcdqiv and the Copernicus Open Access Hub.

**Appendix A: WRF-GHG sensitivity tests**

Sensitivity tests were carried out to identify a model configuration that matches the observations (of $CO_2$, $Ch_4$ and CO) well. We have tested several physical parameterization schemes and anthropogenic fluxes because these elements are essential to accurately simulate tracer concentrations. The initial set of physical parameterization schemes (BASE) was taken from Li et al. (2020) and Dong et al. (2021) as they have shown good model performance for simulating $CO_2$ concentrations in China. Four alternative combinations (A-D) were created by changing the schemes for the longwave and shortwave radiation, planetary boundary layer (PBL) and surface layer physics, leading to 5 different model configurations in total (see Table A1). Remark that there are several more physical parameterization schemes that could have been included in these tests. Nevertheless, a full sensitivity analysis is outside the scope of this study. Thus, we restricted our tests to the most frequently used schemes in the literature and chose the combination that produced satisfactory model simulations without additional optimization.

Further, the following anthropogenic flux inventories were tested: EDGAR GHG v6.0 (for $CO_2$ and $CH_4$, Ferrario et al. (2021)), EDGAR Air Pollutants v5.0 (for CO, Crippa et al. (2019)), CAMS-GLOB-ANT v5.3 (for $CO_2$, $CH_4$ and CO, Granier et al. (2019); Soulie et al. (2023)), PKU v2 (for $CO_2$ and CO, Wang et al. (2013); Zhong et al. (2017)), REAS v3.2.1 (for $CO_2$ and CO, Kurokawa and Ohara (2020)), MEICv3.1 (for $CO_2$ and CO, http://www.meicmodel.org/ ), ODIAC2020b (for $CO_2$, Oda and Maksyuto (2011, 2020); Oda et al. (2018)) and FFDAS v2.2 (for $CO_2$, Asefi-Najafabady et al. (2014)). Monthly fluxes are disaggregated into hourly fluxes using the temporal factors of Crippa et al. (2020), Guevara et al. (2021) and Nassar





et al. (2013). The model code was adapted to include these different anthropogenic emission inventories in separate tracers. As
such, one simulation is sufficient to compare the effect of all inventories.

The five simulations, representing different combinations of physical parameterization schemes and anthropogenic fluxes,
were run over three periods of about 2 weeks spread over the year: 1-17 October 2018, 1-17 February 2019 and 10-25 June
2019. The first 48h were regarded as spin-up and are not taken into account in the analysis.

For each time series the root mean square error (RMSE), mean bias error (BIAS) and Pearson correlation coefficient (CORR)
were calculated. In order to find the most suitable combination of physical parameterization schemes and anthropogenic emis-

sion inventory for all observations at Xianghe, a combined skill score (S) was computed as follows, based on Gbode et al.
(2019):

$$S = (1 - RMSE_{norm}) + (1 - |BIAS_{norm}|) + CORR_{norm}, \tag{A1}$$

where $X_{norm} = \frac{X_i - X_{min}}{X_{max} - X_{min}}$ is the normalized statistical metric. As such, the combination with the highest S will overall have
the lowest RMSE, lowest absolute BIAS and highest CORR. Exact values of the statistical metrics and combined skill scores

for every sensitivity test can be found below in Tables A2, A3, A4, A5 and A6 for the time series of in situ $CO_2$, in situ $CH_4$,
$XCO_2$, $XCH_4$ and XCO, respectively.

Unfortunately, there is not one combination of physical parameterization schemes and anthropogenic flux inventories that
yields optimal scores for all species ($CO_2$, $CH_4$ and CO) across various observation types (surface and column). To identify the
most appropriate model configuration for simulating all observations at the Xianghe site, it is necessary that the chosen physical

parameterization schemes (denoted as test A - D, BASE) show satisfactory skill scores across all five time series. Moreover,
the choice of anthropogenic flux inventory, although potentially varying among species, should yield reasonable score values
for all observation types of the same species. Therefore, the final combination was determined through the following logical
process, where preference was given to the surface data (as it is assumed that the physical schemes will have the highest impact
on these simulations):

– We reject the combination with the worst retults: for each statistical metric, we calculate a threshold derived from the
mean ($\mu$) and standard deviation ($\sigma$) of all occurring values. Combinations in which one or more of the metrics exceed
or fall below these thresholds are excluded from the selection process. Specifically, these combinations must conform to
the following set of equations:

$$\text{CORR} \geq \mu_{\text{CORR}} - \sigma_{\text{CORR}},$$
$$|\text{BIAS}| \leq \mu_{|\text{BIAS}|} + \sigma_{|\text{BIAS}|},$$
$$\text{RMSE} \leq \mu_{\text{RMSE}} + \sigma_{\text{RMSE}} \tag{A2}$$

The combinations that are discarded after this step are indicated with an asterisk ($^*$) behind the inventories name in the
tables below.



- For $CO_2$ and $CH_4$, discard the combinations that are only present in the table of either the surface or either the column data in order to keep only those that are performing good enough on both time series. The combinations that are discarded after this step are highlighted in italic in the tables below.

- From what is left, we see that only combinations with test A, B or C should be considered as those with test D and BASE settings have been discarded for $CH_4$. The choice of physical parameterization option should be the same for all species. When sorting the remaining combinations for $CO_2$ and CO based on $S$ (from the in situ time series for $CO_2$), we find that options with test B and C are superior to those with test A. Finally, a choice has to be made between options with test B and options with test C.

- For both test B and C, we take the emission inventory which has the highest $S$, for $CO_2$ and $CH_4$ based on the in situ time series and for CO based on the column. This leads to the following options:

    - Test B: CAMS-GLOB-ANT for $CH_4$ and $CO_2$; REAS for CO

    - Test C: CAMS-GLOB-ANT for $CH_4$, REAS for $CO_2$ and PKU for CO

- The final choice between these two options is rather arbitrary since certain combinations yield slightly improved results
for one time series but perform less favorably for another, and vice versa. In our study we have chose the combinations with test B.

This approach leads to the settings of test B, together with CAMS-GLOB-ANT v5.3 fluxes for $CO_2$ and $CH_4$ and REAS v3.2.1 (Regional Emission Inventory in Asia) fluxes for CO.

## Appendix B: FLEXPART simulations


The FLEXPART v10.4 model (Pisso et al., 2019) is applied to quantitatively estimate the OH impact on the WRF-GHG $CH_4$ simulation at Xianghe. Table B1 lists the main settings of the FLEXPART model. $CH_4$ particles are released using the FLEXPART backward mode at Xianghe site with and without OH reaction. We release the $CH_4$ particles between 00:00-01:00 and 12:00-13:00 (LT) every day in Oct 2018, Jan 2019, Apr 2019 and July 2019. The release height is set to 0-100 m a.g.l., since
the OH reaction will have a higher impact near the surface where the wind is weaker than at higher altitudes. The backward running duration is set to 3 days, and the backward sensitivities are extending outside of the WRF-GHG d01 boundary. As an example, Fig. B1 shows the spatial distribution of $CH_4$ backward sensitivities for a release at 12:00-13:00 LT on 30 January 2019 including the OH reaction, and Fig. B2 shows the corresponding relative difference in the $CH_4$ backward sensitivities between the simulations with and without OH reaction. Note that FLEXPART v10.4 includes a monthly OH climatology based
on GEOS-Chem simulations (Pisso et al., 2019).



| Test | Flux | CORR | BIAS | RMSE | S |
|------|------|------|------|------|------|
| *B* | *PKU* | *0.67* | *-1.62* | *16.09* | *2.91* |
| **B** | **CAMS** | **0.63** | **-0.12** | **17.50** | **2.81** |
| B | EDGAR | 0.63 | 0.92 | 17.87 | 2.72 |
| *C* | *PKU* | *0.64* | *-3.96* | *16.91* | *2.65* |
| **C** | **REAS** | **0.61** | **-1.19** | **18.88** | **2.58** |
| *A* | *PKU* | *0.63* | *-4.51* | *17.16* | *2.57* |
| *BASE* | *PKU* | *0.61* | *-3.51* | *17.65* | *2.53* |
| *D* | *PKU* | *0.62* | *-4.84* | *17.38* | *2.52* |
| C | FFDAS | 0.58 | -0.92 | 19.12 | 2.50 |
| C | CAMS | 0.59 | -2.77 | 18.06 | 2.49 |
| C | EDGAR | 0.58 | -1.71 | 18.53 | 2.47 |
| D | FFDAS | 0.58 | -1.69 | 19.19 | 2.44 |
| B | REAS | 0.58 | 1.44 | 20.14 | 2.41 |
| B | FFDAS | 0.60 | 2.97 | 20.19 | 2.41 |
| A | CAMS | 0.58 | -3.46 | 18.26 | 2.40 |
| A | EDGAR | 0.57 | -2.46 | 18.58 | 2.40 |
| A | FFDAS | 0.56 | -1.36 | 19.76 | 2.35 |
| C | MEIC | 0.63 | 5.15 | 20.68 | 2.34 |
| D | CAMS | 0.57 | -3.74 | 18.90 | 2.30 |
| D | EDGAR | 0.55 | -2.73 | 19.32 | 2.29 |
| BASE | REAS | 0.55 | -0.29 | 22.00 | 2.27 |
| A | REAS | 0.55 | -1.33 | 21.49 | 2.25 |
| *A* | *MEIC* | *0.59* | *4.60* | *21.84* | *2.20* |
| D | MEIC | 0.58 | 4.02 | 21.86 | 2.19 |
| D | REAS | 0.54 | -1.33 | 22.02 | 2.19 |
| BASE | FFDAS [*] | 0.51 | 0.11 | 21.66 | 2.17 |
| BASE | CAMS [*] | 0.52 | -2.21 | 20.93 | 2.13 |
| BASE | EDGAR [*] | 0.51 | -1.11 | 21.72 | 2.09 |
| B | MEIC [*] | 0.64 | 9.16 | 22.95 | 2.07 |
| BASE | MEIC [*] | 0.57 | 5.94 | 23.34 | 1.99 |
| D | ODIAC [*] | 0.52 | 3.63 | 22.57 | 1.96 |
| C | ODIAC [*] | 0.53 | 5.04 | 22.80 | 1.91 |
| A | ODIAC [*] | 0.49 | 4.56 | 24.46 | 1.69 |
| B | ODIAC [*] | 0.54 | 8.63 | 24.91 | 1.63 |
| BASE | ODIAC [*] | 0.47 | 5.81 | 25.67 | 1.51 |

**Table A2.** Statistical metrics for sensitivity tests, in situ $CO_2$ data at Xianghe. Unit of BIAS and RMSE is ppm. Rows where the inventory name if followed by an asterisk ($*$) indicate those where one or more statistical metrics surpass the thresholds defined in Eq. A2. Rows in italic represent combinations that are rejected due to the $XCO_2$ value falling outside the thresholds. The bold lines represent the final two options as determined by the methodology outlined in Appendix A. **27**



| Test | Flux | CORR | BIAS | RMSE | S |
|------|------|------|------|------|---|
| **C** | **CAMS** | **0.52** | **2.19** | **206.50** | **2.81** |
| C | EDGAR | 0.52 | 19.09 | 208.24 | 2.67 |
| *BASE* | *CAMS* | *0.48* | *3.26* | *213.26* | *2.47* |
| *A* | *CAMS* | *0.45* | *-7.09* | *210.84* | *2.31* |
| *BASE* | *EDGAR* | *0.48* | *22.33* | *216.09* | *2.28* |
| A | EDGAR | 0.46 | 12.50 | 213.59 | 2.27 |
| **B** | **CAMS** | **0.50** | **31.39** | **228.56** | **2.17** |
| B | EDGAR [*] | 0.51 | 52.53 | 237.75 | 1.87 |
| D | EDGAR [*] | 0.41 | 8.83 | 237.26 | 1.70 |
| D | CAMS [*] | 0.39 | -9.19 | 237.31 | 1.60 |

**Table A3.** Same as Table A2 but for in situ $CH_4$. Unit of BIAS and RMSE is ppb.

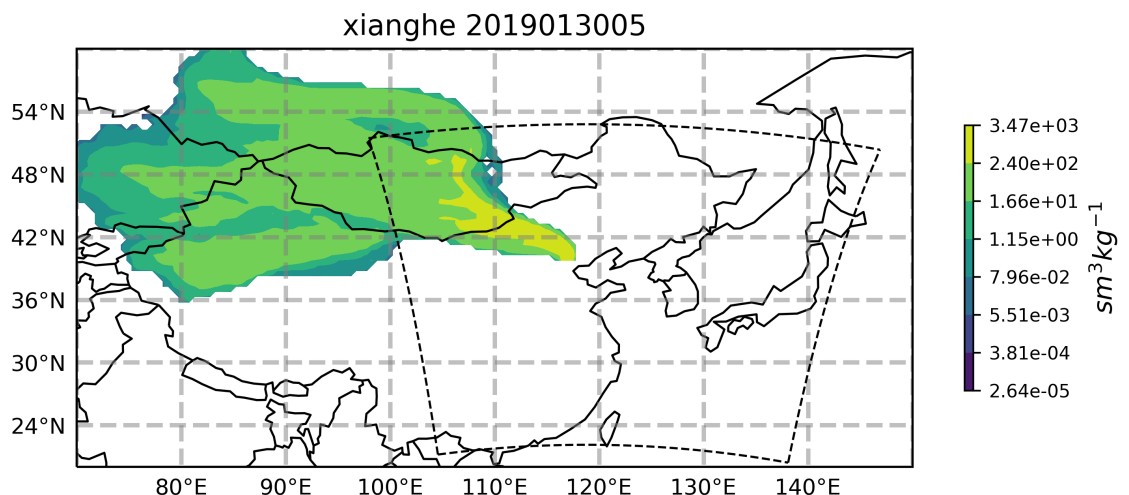

**Figure B1.** The spatial distribution of $CH_4$ backward sensitivities (in $sm^3kg^{-1}$) for a release at 12:00-13:00 LT (which is 04:00-05:00 UTC as indicated in the title) on 30 January 2019 from the FLEXPART simulation including the OH reaction. The location of WRF-GHG d01 is indicated by the dashed line.

*Author contributions.* SC made the model simulations and performed the formal analysis, investigation and visualization. The research was conceptualized by SC, MDM and EM and supervised by MDM and EM. MZ, TW and PW have provided the observational in situ data at Xianghe. BL supported with computing tools to correctly compare the model with TCCON and TROPOMI data. MZ designed and performed the FLEXPART simulations. SC prepared the initial draft of this manuscript while it was reviewed and edited by MZ, BL, TW, MDM, EM and PW.




| Test | Flux | CORR | BIAS | RMSE | S |
|------|------|------|------|------|---|
| D | MEIC | 0.77 | 0.62 | 1.52 | 2.87 |
| C | MEIC | 0.76 | 0.62 | 1.54 | 2.83 |
| BASE | MEIC * | 0.66 | 0.95 | 1.95 | 1.86 |
| *D* | *ODIAC* | *0.78* | *-1.27* | *2.28* | *1.86* |
| *C* | *ODIAC* | *0.79* | *-1.32* | *2.29* | *1.85* |
| A | MEIC * | 0.62 | 0.97 | 2.03 | 1.62 |
| D | FFDAS | 0.80 | -1.60 | 2.43 | 1.58 |
| C | FFDAS | 0.80 | -1.62 | 2.45 | 1.55 |
| *BASE* | *ODIAC* | *0.75* | *-1.36* | *2.47* | *1.54* |
| D | EDGAR | 0.79 | -1.59 | 2.45 | 1.54 |
| *A* | *ODIAC* | *0.74* | *-1.30* | *2.50* | *1.49* |
| *B* | *ODIAC* | *0.72* | *-1.23* | *2.53* | *1.47* |
| C | EDGAR | 0.77 | -1.57 | 2.49 | 1.45 |
| D | CAMS | 0.79 | -1.70 | 2.52 | 1.38 |
| B | EDGAR | 0.76 | -1.54 | 2.55 | 1.38 |
| B | FFDAS | 0.75 | -1.52 | 2.58 | 1.33 |
| **C** | **REAS** | **0.80** | **-1.81** | **2.56** | **1.33** |
| *BASE* | *FFDAS* | *0.77* | *-1.65* | *2.58* | *1.32* |
| D | REAS | 0.79 | -1.80 | 2.56 | 1.32 |
| *BASE* | *EDGAR* | *0.77* | *-1.64* | *2.59* | *1.31* |
| C | CAMS | 0.77 | -1.68 | 2.57 | 1.30 |
| A | FFDAS | 0.76 | -1.62 | 2.60 | 1.28 |
| **B** | **CAMS** | **0.75** | **-1.64** | **2.63** | **1.22** |
| *BASE* | *CAMS* | *0.76* | *-1.74* | *2.66* | *1.15* |
| D | PKU * | 0.80 | -1.98 | 2.67 | 1.13 |
| C | PKU * | 0.80 | -2.00 | 2.67 | 1.13 |
| B | REAS | 0.75 | -1.71 | 2.69 | 1.10 |
| A | EDGAR | 0.72 | -1.54 | 2.71 | 1.09 |
| BASE | REAS | 0.76 | -1.85 | 2.73 | 1.03 |
| A | CAMS | 0.72 | -1.65 | 2.76 | 0.98 |
| A | REAS | 0.75 | -1.84 | 2.75 | 0.97 |
| B | MEIC * | 0.55 | 1.25 | 2.30 | 0.95 |
| B | PKU * | 0.76 | -1.91 | 2.77 | 0.94 |
| BASE | PKU * | 0.77 | -2.02 | 2.81 | 0.87 |
| A | PKU * | 0.76 | -2.00 | 2.82 | 0.84 |

**Table A4.** Same as Table A2 but for $XCO_2$.



| Test | Flux | CORR | BIAS | RMSE | S |
|------|------|------|------|------|------|
| **B** | **CAMS** | **0.69** | **-0.79** | **20.53** | **2.94** |
| *B* | *EDGAR* | *0.69* | *0.65* | *20.94* | *2.62* |
| C | EDGAR | 0.67 | -0.96 | 21.24 | 1.73 |
| *D* | *EDGAR* | *0.66* | *-0.80* | *21.45* | *1.47* |
| **C** | **CAMS** | **0.67** | **-2.16** | **21.31** | **1.12** |
| A | EDGAR | 0.65 | -1.17 | 21.72 | 0.86 |
| BASE | EDGAR* | 0.65 | -1.66 | 21.76 | 0.59 |
| D | CAMS* | 0.65 | -2.09 | 21.75 | 0.55 |
| A | CAMS* | 0.65 | -2.75 | 21.45 | 0.37 |
| BASE | CAMS* | 0.65 | -3.03 | 21.42 | 0.34 |

**Table A5.** Same as Table A2 but for $XCH_4$. Unit of BIAS and RMSE is ppb.

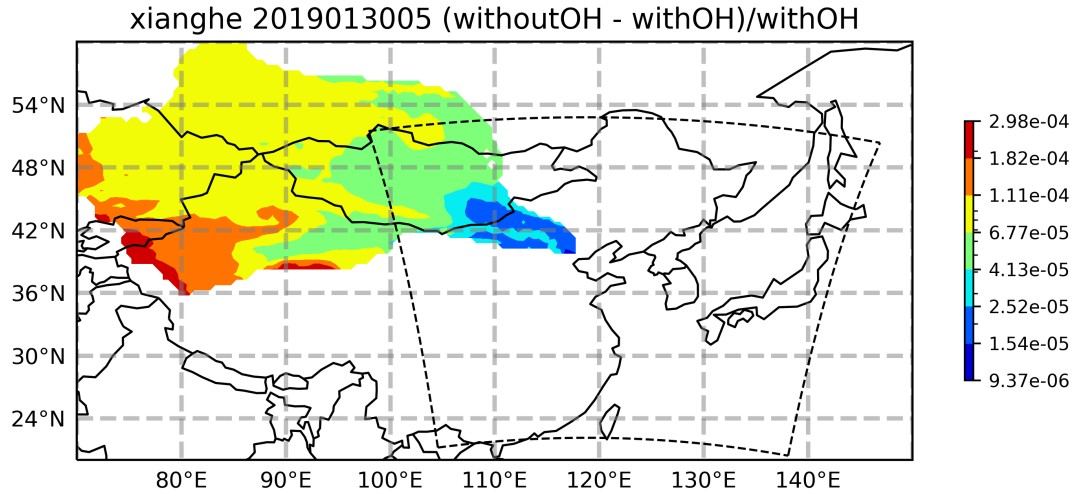

**Figure B2.** Relative difference in the $CH_4$ backward sensitivities between simulations with and without OH reaction. The location of WRF-GHG d01 is indicated by the dashed line.

*Competing interests.* The contact authors have declared that neither they nor their co-authors have any competing interests.

*Disclaimer.* The results contain modified Copernicus Climate Change Service information 2022. Neither the European Commission nor ECMWF is responsible for any use that may be made of the Copernicus information or data it contains.



| Test | Flux | CORR | BIAS | RMSE | S |
|------|------|------|------|------|---|
| **B** | **REAS** | **0.78** | **-3.99** | **30.25** | **2.96** |
| B | PKU | 0.78 | -5.38 | 30.32 | 2.94 |
| D | REAS | 0.76 | -5.32 | 31.54 | 2.83 |
| BASE | REAS | 0.77 | -7.12 | 31.39 | 2.81 |
| **C** | **PKU** | **0.76** | **-6.88** | **31.73** | **2.79** |
| D | PKU | 0.76 | -6.82 | 31.75 | 2.79 |
| C | REAS | 0.75 | -5.53 | 32.00 | 2.79 |
| BASE | PKU | 0.77 | -7.85 | 31.74 | 2.77 |
| A | REAS | 0.75 | -7.15 | 32.35 | 2.72 |
| A | PKU | 0.75 | -7.51 | 32.67 | 2.71 |
| B | CAMS | 0.68 | -24.21 | 43.48 | 1.70 |
| BASE | CAMS | 0.66 | -25.16 | 44.35 | 1.61 |
| D | CAMS | 0.64 | -24.34 | 44.34 | 1.57 |
| BASE | EDGAR [*] | 0.52 | 3.27 | 57.84 | 1.45 |
| A | CAMS | 0.59 | -23.19 | 44.78 | 1.44 |
| C | CAMS | 0.60 | -23.77 | 45.07 | 1.43 |
| B | EDGAR [*] | 0.53 | 5.90 | 59.57 | 1.34 |
| A | EDGAR [*] | 0.50 | 6.27 | 62.83 | 1.16 |
| D | MEIC [*] | 0.65 | -37.13 | 49.72 | 1.05 |
| C | MEIC [*] | 0.61 | -37.10 | 50.26 | 0.94 |
| B | MEIC [*] | 0.53 | -30.94 | 47.80 | 0.93 |
| C | EDGAR [*] | 0.46 | 8.30 | 67.22 | 0.87 |
| D | EDGAR [*] | 0.47 | 8.66 | 68.01 | 0.86 |
| BASE | MEIC [*] | 0.55 | -34.80 | 49.89 | 0.82 |
| A | MEIC [*] | 0.52 | -34.49 | 50.35 | 0.72 |

**Table A6.** Same as Table A2 but for XCO. Unit of BIAS and RMSE is ppb.



| Parameter | Settings |
|---|---|
| Release location | $\pm 0.1^\circ$ around Xianghe site |
| Release height | 0 - 100 m.a.g.l |
| Release time | 00:00-01:00 and 12:00-13:00 (LT) every day in Oct 2018, Jan 2019, Apr 2019 and July 2019 |
| Number of backward running days | 3 |
| Number of releasing particles | 20 000 |
| OH reaction | On and off |
| Meteorological data | NCEP CFSv2 with $0.5^\circ \times 0.5^\circ$ horizontal resolution and 64 vertical levels (Saha et al., 2014) |

**Table B1.** The main settings of FLEXPART model run with a $CH_4$ tracer

*Acknowledgements.* We would like to thank all staff at the Xianghe site for operating the FTIR and PICARRO measurements. This work
is supported by the National Natural Science Foundation of China (No. 42205140; 41975035). Emmanuel Mahieu is a senior research
associate with the F.R.S.-FNRS. The authors acknowledge all providers of observational data and emission inventories. We thank the IT
team at BIRA-IASB for their support on data storage and HPC maintenance. Christophe Gerbig, Roberto Kretschmer, and Thomas Koch
(MPI BGC) are thanked for distributing the VPRM preprocessor code. Finally, we are grateful for fruitful discussions with Jean-François
Müller (BIRA-IASB) and Bernard Heinesch (ULiège).



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
