# Peer review of "A WRF-Chem study of the greenhouse gas column and in situ surface concentrations observed at Xianghe, China. Part 1: Methane $(CH_4)$"

_EGUsphere, 2024_

## Author Response (AR1)

**Author comments on " WRF-Chem study of the greenhouse gas column and in situ surface concentrations observed at Xianghe, China. Part 1: Methane (CH$_4$)"**

We wish to thank the reviewers for their careful reading of our manuscript and their constructive comments. We will address each of the comments below. The reviewer comments are written in black, while our author comments are in blue. Citations from the manuscript are in cyan, where modifications are shown in *italic*.

**1   reviewer 1**

**Specific Comments:**
Page 3, line 85: In Yang et al., 2021 it is reported that the WMO X2007 calibration scale was used, did you switch to the "new" WMO CO2 X2019 scale?
We appreciate the reviewer's comment. Indeed, the in situ CO$_2$ observations follow the WMO X2007 calibration scale, while the CH$_4$ observations use the X2004A scale. For CO$_2$, the data can be converted to the WMO X2019 scale using the transformation $X2019 = 1.00079 \cdot X2007 - 0.142$. This conversion will be applied in the second part of our work, which will be focused on CO$_2$. However, the current study discusses results for CH$_4$, where the X2004A scale remains up-to-date. We believe that addressing the CO$_2$ calibration scale is not necessary in this part of the work.

Pag. 4, line 92: What do you mean by "a quality filter of 1.0 was applied"?
Thank you for pointing out this issue. It seems there has been a mistake in writing. Instead it should be written: *a quality filter of 0.5*. This refers to the quality assurance value (*qa_value*) as described in the Product Readme File (PRF) for the Sentinel 5 Precursor Tropospheric Monitoring Instrument (S5P/TROPOMI) Methane (CH4) total column level 2 data product. For the highest quality data, it is recommended to apply this quality filter of 0.5. We have adjusted this accordingly in the manuscript.

Table 2. Why did you decide to merge the very different emissions sectors of residential and waste? Does, also in light of your discussion on page 13, the separation of these sectors allow for refinements in the attribution of differences between simulation and observations?
In the initial setup, sector definitions were kept uniform across all species to maintain consistency, as the CH$_4$ simulations presented in this manuscript are part of a larger set of simulations that also includes CO$_2$ and CO. Since waste emissions are only relevant for CH$_4$, a separate waste tracer was not introduced at the time to avoid adding a sector exclusive to one species. Additionally, the potential significance of waste emissions was underestimated at that stage.
In retrospect, distinguishing waste from residential emissions for CH$_4$ could have provided a more detailed attribution of source contributions by providing quantitative values of their relative importance. However, given the substantial computational resources required for these one-year simulations, modifying the tracer setup at this point is unfortunately not feasible. We would recommend however, to distinguish waste and residential tracers in future simulations, and we will implement this approach ourselves in subsequent studies. That said, the key conclusions of our analysis would remain largely unaffected, as the focus is on fluxes rather than tracers. The discussion on page 13 explores potential causes for the model's summer underestimation by exploring in more detail the monthly flux variations as they are reported in the emission inventories. Finally, we don't expect to see large differences in the relative surface and column contributions between the

[Figure]

Figure A: Map of annual CH$_4$ emissions (Tg) in CAMS-GLOB-ANT inventory for the (a) solid waste and waste water and (b) residential, commercial and other combustion sectors. Note that both panels have different colorbar limits. The location of the Xianghe site is indicated by a black cross.

Residential and Waste sectors if we would have separated them since the spatial distribution of both fluxes is very similar (see Fig. A). The main added value would be to have more detail quantitatively.

Table 3 & Figure 2: Looking at the wide range of deviations (model - observations) reported in Figure 2d, ranging from -1000 ppb to +1000 ppb, I have the feeling that the statistics reported in Table 3 give an overly "optimistic" view of the discrepancies with in-situ observations. I would suggest to also report the main quantiles of the model - observations deviations. However, looking at Fig. 2, it seems that the agreement would increase over longer time scales (i.e. day-to-day).

It is true that there is a large range in the hourly values of the model - observation differences, partly due to the large variability at this temporal scale. To provide a more complete view on the error distribution, we propose to add a bar plot of its distribution (see Fig. B) next to the table of statistics (Table 2 in the preprint, or Table 1 here), accompanied with the following sentence in the manuscript: *"To further illustrate the distribution of the differences between model and observations, a histogram is given in Fig. B."*

Table 1: Statistics of the model-data comparison of the ground-based CH$_4$ observations at the Xianghe site from 1 September 2018 until 1 September 2019. We present the mean bias error (BIAS), root mean square error (RMSE) and Pearson correlation coefficient (CORR). The mean bias error and root mean square error are given in ppb. For in situ observations, the data is split in afternoon (13-18 LT) and night (3-8 LT) hours.

|  | in situ CH$_4$ (afternoon) | in situ CH$_4$ (night) | XCH$_4$ |
|---|---|---|---|
| BIAS | 14.22 | 12.68 | -3.03 |
| RMSE | 159.74 | 334.06 | 23.96 |
| CORR | 0.66 | 0.42 | 0.56 |

Pag. 13, line 259: What fugitive CH4 emissions are associated with coal transport?
Indeed, this sentence is strictly speaking incorrect as fugitive emissions are mainly linked to gas and oil processes and not coal. We propose to omit the 'fugitive' adjective and change the sentence to: *"emissions from the extraction, processing, storage and transport of coal, oil and natural gas.'*

Pag. 14, line 14: I think it's rather challenging to get hints about misrepresentation of emissions in WRF-CHEM by directly comparing CH$_4$ fields with TROPOMI measurements as done in Figure 6. This approach misses any inaccuracies in atmospheric transport and chemistry. I would suggest that these points are highlighted in the discussion and that direct comparisons with TROPOMI are treated with more caution (also

[Figure]

Figure B: Distribution of difference between model and observations for (a) XCH$_4$ and (b) in situ CH$_4$ (ppm), using hourly data points. The red line indicates the median difference, while the blue lines are the 25% and 75% quantiles.

in Section 3.5).
We understand the concern that the direct comparison with TROPOMI in Figure 6 is influenced by potential inaccuracies in atmospheric transport and the absence of chemistry within WRF-GHG. We acknowledge that transport uncertainties are relevant for TROPOMI comparisons, as well as for comparisons with TCCON data at Xianghe. However, for this study, the simulations are re-initialized with ERA5 reanalysis every 30h and we operate under the assumption that the WRF-GHG simulations do not exhibit systematic errors in atmospheric transport. As a result of a comment of the second reviewer, this was explicitly added to the manuscript in section 3.1 (see below). It is necessary to consider this assumption when interpreting the results. It should however also be noted that the comparison in Figure 6 is made on seasonal averages, which will mitigate the impact of random transport errors, as illustrated in Figure 12a-b. Further, we agree that the exclusion of atmospheric chemistry in our simulations is a limitation, which has been discussed at various points in the manuscript. To address the reviewer's concern directly within the manuscript, we propose to emphasize the challenges of this comparison in the discussion of Figure 6 by adding the following sentence: *"Note that this comparison with TROPOMI should be interpreted with caution, as WRF-GHG does not account for atmospheric chemistry and is assumed to not show systematic transport errors. The primary aim here is to identify suggestive spatial patterns of potential emission inaccuracies."*
Finally, we also propose to modify the discussion in section 3.5, as will be elaborated in the corresponding comment below.

Section 3.4.1: Reading the first few words of this section, I had the impression that the analysis was also based on FTIR data, but this was not actually the case. I think that showing the same analysis also for FTIR data, as reported in Figure 7 for WRF-CHEM, may provide more guidance in assessing the agreement between observations and simulation, as well as the role of synoptic transport in the column CH$_4$ observations.
In section 3.4.1. we aim to find a correlation between tracer enhancements and upper wind direction. Since the WRF-GHG simulations allow a distinction between contributions from within the model domain (tracers) and from outside (background), it is more convenient to perform this analysis on the model data. We agree that additionally including the FTIR data may provide a more complete image. We therefore propose to add a second panel to Figure 7 in the manuscript, see Fig. C. Panel (b) shows the distribution of the daily mean observed XCH$_4$ values (ppb) per (simulated) wind direction. We observe a significant difference for the FTIR data as well (see the very low p-value in the title), however it is slightly smaller than the difference between the modeled enhancements because the observations are a mixture of background and tracers.

[Figure]

Figure C: Distribution of the daily mean (a) simulated column tracers above the background and (b) observed XCH$_4$ per 800 hPa wind direction category. NW is for winds with an angle of 292.5 to 337.5 ° from north, while SW represents the angles between 202.5 and 247.5 °. There are 72 days with NW winds and 33 days with SW winds. The colored boxes indicate the range between the first and third quartile, while the thick solid line is the median. Outliers (values that are 1.5 times the interquartile range above (below) the third (first) quartile) are shown by black dots. The p-values of the corresponding non-parametric Mann-Whitney U tests are given in the title.

Page 18, line 387: How was the 45 ppb threshold for XCO defined?

The aim of establishing a threshold was to define a criterion for filtering out days characterized by significant pollution. As demonstrated in Section 3.4.1, these events are generally associated with upper-level winds originating from the southwest. This finding aligns with the results of Yang et al., 2020, which indicated a strong correlation between elevated levels of XCO$_2$, XCH$_4$, and XCO, linked to pollution originating from the North China Plain. We generated a boxplot of XCO enhancements (Fig. D), analogous to the one presented in Fig. C, which reveals that on days with a daily mean 800 hPa wind direction from the southwest, the average XCO enhancement typically exceeded 45 ppb (with the 25th percentile at 47.48 ppb). Conversely, such elevated enhancements were rarely observed on days with more northerly wind directions, which are indicative of cleaner air masses. We chose to focus on XCO enhancement as our primary indicator due to the more robust correlation observed between the trace gases compared to the correlation with wind direction. Finally, the threshold of 45 ppb was pragmatically selected based on the distribution of values in Fig. D to provide a practical filter for distinguishing between polluted and relatively clean days.

Figure 9: I suggest adding a wind rose to Figure 9 for the in situ data. This would allow a direct assessment of local emissions against the observed data. To save space, plates (f) can be moved to the supplementary material.

The panels in Figure 9 present an alternative visualization of the correlation between wind and tracer concentration data, offering a different perspective compared to a conventional wind rose. Typically, a wind rose illustrates the frequency of wind occurrence from various directions using bars, the length of which corresponds to the prevalence of each direction. These bars can be divided into concentration bins, indicating the frequency of specific concentration values within each wind direction. A separate panel might be added to display wind speed distribution per direction. In contrast, the current figure shows colored bins organized by wind speed and direction, where the color intensity of each bin represents the mean tracer concentration for

[Figure]

Figure D: Similar as Figure 7 in manuscript or Fig. Ca but for CO. Distribution of the daily mean simulated column tracers for CO above the background, per 800 hPa wind direction category.

that specific wind speed and direction on a continuous scale. As this approach does not explicitly show the frequency of particular wind directions (and associated wind speeds), we had included supplementary panels (f,l) displaying the bin counts.

However, we acknowledge that these alternative wind rose representations may be less intuitive than the standard format, despite presenting equivalent information. Therefore, we propose to revise the figure to incorporate conventional wind rose panels (as shown in Fig E). Furthermore, in response to the referee's suggestion, we recommend relocating the panels illustrating the wind speed frequency per direction to the appendix (as depicted in Fig F).

Page 19, line 396: For wetlands there is also an apparent contribution from E during the night.

Indeed, as the major wetland sources in the region of Xianghe are located to the east - southeast, we find larger wetland tracer concentrations for that direction. This is especially visible at night (Fig. Ej) since east winds are very rare during the day (Fig. Fa). We propose to be more precise in the manuscript by stating: "In general, we see that wind directions with the largest enhancements correspond with the largest sources nearby: ... and *east-southeast* for wetlands."

Section 3.5: Based on the comparison between annual TROPOMI and WRF-CHEM $XCH_4$ fields (see my previous concern about this approach), this section concludes that the contribution from coal mines near Tangshan may be overestimated by the CAMS-GLOB-ANT emission fluxes. However, looking at Figure 10, there are other sectors (energy, industry) contributing to this emission hotspot. How can you say that only emissions from coal mines contribute to the higher XCH4 values of WRF-GHG? Furthermore, the deviations of TROPOMI vs. WRF-GHG were about the same order of magnitude as the biases reported in Section 2.2 for TROPOMI, so are they really significant?

Firstly, the hypothesis that coal mine emissions contribute to an overestimation of in situ $CH_4$ concentrations is supported by the dominance of the Energy sector emissions, which are at least five times greater than any other sector and up to twenty times larger than Industry emissions (Fig. 10; note varying color scales across panels). Furthermore, the more detailed subdivision of Energy sector emissions within CAMS-GLOB-ANT, indicates that the identified hotspots are specifically attributed to coal mining activities, rather than oil or gas extraction. Consequently, any inaccuracies associated with the Energy tracer are highly likely linked to the representation of emissions from the coal sector.

Mean CH$_4$ tracer (ppb) per wind speed and direction

[Figure]

Figure E: Proposed alternative for Figure 9 in manuscript.

[Figure]

Figure F: Figure for in appendix showing the frequency of wind speed and wind direction

For the second comment, we recognize that directly comparing TROPOMI and WRF-GHG data can be tricky. To eliminate the overall bias between the two datasets, we focused on the difference in their average errors across specific regions. First, we defined a broader area (BCK) around the target emission source with minimal other emissions. Within BCK, we identified a smaller core area (SRC) specifically containing the coal mine source from the emission inventory. Our approach was to compare the average error within SRC to the average error in the surrounding area (BCK excluding SRC). By looking at this difference, we aimed to assess if the error around the coal mine source is more pronounced than its surroundings, suggesting overestimation of the emissions. While a statistical test focusing only on the variability within BCK and SRC initially suggested a significant difference in bias, this test didn't account for the fact that the bias isn't uniform across the larger area. Indeed, other studies indicate that TROPOMI's bias can vary spatially: the validation report of the S5P Mission Performance Centre (MPC) presents a dispersion of 0.7% for the TROPOMI bias among different FTIR sites. Considering this spatial variability of the bias, the 3 ppb difference we found between SRC and the surrounding area (BCK-SRC) is likely not statistically meaningful. We therefore propose to adjust the relevant paragraph in the manuscript to the following:

"We observe slightly elevated $XCH_4$ values near the coal mines of Tangshan in both the WRF-GHG and TROPOMI maps. To isolate the potential model bias over these sources, we defined a surrounding background region (39.3-40 °N, 117.8-118.8 °E), characterized by the absence of major $CH_4$ emissions, and a source region (39.45-39.8 °N, 118.15-118.6 °E) encompassing the concentrated coal mine sources from CAMS-GLOB-ANT. The mean difference (WRF-GHG - TROPOMI) was 8.87 ± 2.22 ppb in the background region and 11.85 ± 3.24 ppb over the source region, suggesting a potentially larger model overestimation near the coal mines. *However, when considering the reported spatial variability of the TROPOMI bias, this difference of approximately 3 ppb is not statistically significant. Validation studies by the S5P Mission Performance Centre (MPC) indicate a bias dispersion of 0.7% across different FTIR sites, which translates to a potential bias variation exceeding 13 ppb. Therefore, based on this TROPOMI comparison alone, we cannot definitively conclude that the CAMS-GLOB-ANT emissions from these sources are overestimated. Additional in situ $CH_4$ measurements in the immediate Tangshan area are crucial for a more robust assessment.*"

Consequently, the corresponding part in Section 4: Conclusions would be adjusted: "Furthermore, comparisons between simulated and observed $CH_4$ concentrations near the surface, suggest an overestimation of coal mine emissions near Tangshan in the emission inventory of CAMS-GLOB-ANT. *However, this hypothesis remains unconfirmed due to the averaging effect in the column measurements, the relatively low emission strength and the reported accuracy of TROPOMI $XCH_4$.*"

Page 21, line 414: "This suggests ... sources". Are you referring to TROPOMI or WRF-CHEM?
In this specific sentence, we are referring to TROPOMI $XCH_4$ maps. The comparison of the CAMS-GLOB-ANT emission flux map with the simulated annual mean $XCH_4$ map from WRF-GHG (Fig. 12a-b) demonstrates that yearly averaged $XCH_4$ fields can effectively highlight the strongest emission sources. Given that the WRF-Chem annual mean $XCH_4$ map was generated using only pixels with corresponding TROPOMI observations, we argue that this principle likely extends to TROPOMI $XCH_4$ maps as well. The annual averaging process minimizes the impact of meteorological patterns and random errors. However, the retrieval of individual emission source strengths from TROPOMI $XCH_4$ maps remains challenging, as it is primarily feasible for the most substantial sources due to the instrument's accuracy limitations (see also discussion of previous comment).

**2   reviewer 2**

**Specific Comments:**
Page 5, Table 1: Was the cumulus parameterization activated also in the last nested grid (D03)? Since it is generally not recommended for high-resolution domains, clarification on this decision would be valuable.
Indeed, cumulus parameterization schemes are typically only recommended for model simulations with a coarse horizontal resolution of 10 km or more. At finer resolutions, the so-called 'gray zone' emerges, where convective processes are only partially resolved, introducing uncertainties in using parameterization schemes. Therefore, we applied a cumulus parameterization scheme only in the parent domain (d01), which has a resolution of 27 km, and not in d02 or d03 with resolutions of 9 km and 3 km, respectively. We clarify this in Table 1 by stating "*Cumulus (only in d01)*" in the first column and by adding the following sentence in

 *"Remark that the cumulus parameterization scheme was only applied in the outermost domain (d01)."*

Page 5, Line 113: How was the interpolation of the CAMS EGG4 global product performed for the model? It would be beneficial to include this information. Additionally, was any other boundary condition product tested besides the CAMS global reanalysis for greenhouse gases (EGG4)?

The CAMS data was regridded to the model domain using the *remapcon* function of the CDO software (Climate Data Operators, Schulzweida, 2020), which performs a first order conservative remapping. We added the following in the text (at line 114): *"The CAMS reanalysis fields are remapped with mass conservation to the WRF-GHG domains using the CDO software (Climate Data Operators, Schulzweida, 2020)."*
No other boundary condition product was tested.

Page 9, Figure 2: Why do the differences between the model and observations exhibit distinct patterns when comparing $XCH_4$ column values and in situ measurements? In September, October, and November 2018, the model underestimates $XCH_4$ column concentrations, whereas the in situ values do not show the same underestimation. However, for the rest of the period, the two comparisons exhibit a similar pattern. Discussing this discrepancy would strengthen the analysis.

Indeed, a distinct discrepancy exists in the model-observation biases when comparing $XCH_4$ column and in situ $CH_4$ measurements. However, this distinct pattern is primarily evident in September 2018, as illustrated in Figure G, providing a closer look at the monthly deviations. During this month, the model shows an underestimation of $XCH_4$ column concentrations, while simultaneously overestimating in situ $CH_4$ values.
We hypothesize that this divergence is linked to agricultural emission sources. As detailed in Section 3.2 and Figure 4, the tracer associated with agricultural fluxes significantly influences the total $CH_4$ signal at the Xianghe site in September 2018. More specifically, the agricultural tracer's mean contribution exceeds the sum of all other tracers for both column measurements (23.41 ppb for agriculture vs. 21.94 ppb for sum of other tracers) and in situ $CH_4$ measurements (238.25 ppb vs. 156.78 ppb), but with a more pronounced effect near the surface.
This pronounced influence at the surface is likely related to the spatial distribution of agricultural emission sources within the CAMS-GLOB-ANT inventory and the different sensitivities of both measurement types. As depicted in Figure 5a, a localized high-emission area is present in the region immediately north of Xianghe. Our interpretation is that these emissions are overestimated in September, leading to an overestimation of in situ $CH_4$ at Xianghe. Conversely, since column observations integrate over a much larger spatial domain, this assumed overestimation in localized fluxes would have a less pronounced impact on the corresponding tracer and total simulated fields. As noted in Section 3.1, background $CH_4$ concentrations contribute significantly to the total simulated signal, especially for $XCH_4$. The hypothesis of overestimated agriculture emissions in September, can further be linked to the discussion in section 3.3 where the temporal variation of the agriculture sector is debated. We propose to make the following adjustments in the manuscript:

In section 3.1 where the bias is initially described: *"Moreover, the same seasonal pattern is found in the time series of the differences for the in situ data, except in September (Fig. 2d)."*
The discussion on emissions of the agriculture sector when investigating the causes of the seasonal $CH_4$ bias in section 3.3 would become the followning:
*"As presented in Table 2, the agricultural sector is comprised of three subsectors: soils (this is mainly rice cultivation), agricultural waste burning, and livestock (manure management and enteric fermentation). In China, rice cultivation plays a vital role but is predominantly concentrated in regions south of 35°N. In CAMS-GLOB-ANT, the most important agriculture subsector in the region of the Xianghe site is livestock. According to the emission inventory, livestock emissions in the wide region around Xianghe peak in September and reach their lowest levels in March and April. Unfortunately, the source of these monthly variations in $CH_4$ emissions within the inventory is unclear, as the accompanying data set of temporal factors, CAMS-GLOB-TEMPO (Guevara et al., 2021), references constant factors for $CH_4$ emissions from agricultural sources. Previous research by Maasakkers et al., 2016 suggests that emissions from manure management often correlate with air temperature, with higher emissions during warmer months (May to September in this case) and lower emissions during colder months (December to February). If the true seasonality of agricultural emissions around Xianghe is indeed temperature-driven, it implies that the current inventory underestimates*

[Figure]

Figure G: Monthly mean difference between model and observations for (a) XCH$_4$ and (b) in situ CH$_4$ (in ppb).

emissions during spring and summer (May to August) and overestimates them in winter, as it shows a peak only in September and a minimum in spring (March-April) rather than in winter. This discrepancy in the seasonality of emissions could explain the seasonal bias observed in our CH4 simulations, pointing to inaccuracies in the representation of agricultural emissions. *This is further evidenced by the contrasting model biases observed in September between near-surface and column-averaged CH$_4$ (Fig.2 c-d). Incorrect temporal variation of agricultural emissions implies an overestimation of the emission in September, where the current peak values are. Considering that in situ observations are more sensitive to local sources compared to column measurements, which represent a larger spatial average, and given the presence of a localized high-emission area immediately north of Xianghe (Fig. 5a), this overestimation in September would more likely lead to an overestimation of in situ CH$_4$ than XCH$_4$.*"

How well does the model perform in simulating meteorological fields? Including an assessment of statistical performance would be valuable.

Due to the absence of qualitative observational meteorological data at the Xianghe site, a direct analysis of local meteorological parameters was not included in this study. Furthermore, readily available regional meteorological datasets for comparative analysis were also lacking. Consequently, we explored the publicly accessible Global Hourly - Integrated Surface Database (ISD) from the National Centers for Environmental Information (NCEI) to assess the meteorological performance of WRF-GHG. This comprehensive dataset encompasses hourly surface observations collected from diverse global sources. Within our WRF-GHG model domain d03 (3 km horizontal resolution) and the simulation period spanning 1 September 2018 to 1 September 2019, meteorological observations were available from 14 synoptic stations (summarized in Table 2). These selected stations report various parameters, including wind speed and direction, temperature, dew point, cloud information, and visibility, at hourly or 3-hourly intervals. For the purpose of a model performance evaluation, we extracted simulated 2-m temperature and 10-m wind speed and direction from the WRF-GHG output corresponding to the grid cells where these meteorological observations were recorded. For sites reporting data at 3-hour intervals, comparisons were made with a 3-hour rolling mean of the simulated data. For hourly reporting sites, a direct comparison with the hourly model output was conducted. Only data points meeting specific quality control criteria (quality flags of 0, 1, 4, 5, or 9) were included in the analysis. An overview of the model performance against these meteorological observations is presented in Figure H.

The WRF-GHG model demonstrates a reasonable capability in simulating surface meteorological fields. Across domain d03 and the entire simulation period, a strong correlation of 0.98 and a small mean bias error of 0.82 °C is observed for near-surface temperature. For wind speed, a moderate correlation of 0.56 is found, accompanied by a slight model overestimation of 0.84 ms$^{-1}$ on average, and a root mean square error (RMSE) of 2.21 ms$^{-1}$. The 10-m wind direction simulated by WRF-GHG exhibits a mean bias error of approximately 48° relative to the surface observations. It is important to note, however, the relatively low precision of the available observations: 0.1 °C for temperature, 1 ms$^{-1}$ for wind speed, and 10-30° (varying by site) for wind direction. Considering these limitations in observational precision and the absence of other qualitative meteorological measurements, we conclude that the model performance for these parameters is

[Figure]

Figure H: Error distribution for (a) 2-m temperature, (b) 10-m wind speed and (c) 10-m wind direction between WRF-GHG simulations in d03 and 14 synoptic weather stations within d03.

acceptable. We propose to add a concise summary of this meteorological model evaluation in the manuscript in the beginning of section 3.1:

*"Due to the absence of qualitative meteorological observations at the Xianghe site, the simulated near-surface temperature and 10-m wind fields within domain d03 were evaluated against the publicly available Global Hourly - Integrated Surface Database (ISD) from the National Centers for Environmental Information (NCEI) to assess model performance for key meteorological parameters. Considering the reported precision of the observational data, the analysis reveals that the WRF-GHG model adequately captures the primary surface meteorological conditions within the study domain and period. We therefore assume that significant systematic errors in the simulated transport are unlikely."*

Page 14, Line 312: The methodology for the simulations using the FLEXible PARTicle dispersion model is only mentioned in the Appendix. A brief explanation should be included in the main text for clarity.

Thank you for this suggestion. We modified the relevant part of the section to the following: "Finally, we used backward simulations with the FLEXible PARTicle dispersion model (FLEXPART) v10.4 (Pisso et al., 2019) to evaluate the impact of the OH sink on $CH_4$ concentrations at Xianghe. *$CH_4$ particles were released near the surface at the Xianghe site using the FLEXPART backward mode with and without OH reaction, at different times throughout the day and year.* More specific details of the model configuration and simulations are provided in Appendix B. By comparing simulations that include or exclude the chemical reaction with OH, we estimated its influence. The results indicate ..."

**Minor Comments:**

Figure 6: It would be helpful to indicate the location of Xianghe in the plots.
Indeed, we have added a black cross in all panels of Figure 6.

Why are the table captions generally placed at the bottom? Consistency in formatting should be considered.
The captions of all figures and tables in the manuscript are placed at the bottom, and are therefore already consistent within this work. However, we assume the referee is hinting at the fact that in most publications, the caption for tables is placed on top, while it is placed at the bottom for figures. We will adjust the captions in the manuscript accordingly.

Adding background grid lines to all figures with concentration axes would facilitate comparisons between the model and observations.
Thank you for the suggestion. We will add background lines to all figures with concentration axes.

Page 11, Figure 4: The color palette for the "Residential & Waste" and "Agriculture" sectors should be

Table 2: List of sites with meteorological surface observations within WRF-GHG d03 and between 2018-09-01 and 2019-09-01, from the NCEI's Global Hourly - Integrated Surface Database. The last column indicates the type of geophysical surface observation: FM-12 represents "SYNOP Report of surface observation form a fixed land station" while FM-15 corresponds with "METAR Aviation routine weather report".

| Site | Latitude | Longitude | Elevation (m) | Report type |
|---|---|---|---|---|
| ZHENGDING, CH | 38.28 | 114.70 | 71.01 | FM-15 |
| CHENGDE, CH | 40.97 | 117.92 | 423 | FM-12 |
| BEIJING CAPITAL INT. AIRPORT, CH | 40.08 | 116.58 | 35.35 | FM-12 |
| BEIJING CAPITAL INT. AIRPORT, CH | 40.08 | 116.58 | 35.35 | FM-15 |
| TIANJIN, CH | 39.10 | 117.17 | 5 | FM-12 |
| TANGSHAN, CH | 39.65 | 118.10 | 29 | FM-12 |
| ZHANGJIAKOU, CH | 40.78 | 114.88 | 726 | FM-12 |
| FENGNING, CH | 41.20 | 116.63 | 661 | FM-12 |
| POTOU, CH | 38.08 | 116.55 | 13 | FM-12 |
| BINHAI, CH | 39.12 | 117.35 | 3.04 | FM-15 |
| LETING, CH | 39.43 | 118.90 | 12 | FM-12 |
| HUAILAI, CH | 40.42 | 115.50 | 538 | FM-12 |
| BAODING, CH | 38.73 | 115.48 | 17 | FM-12 |
| QINGLONG, CH | 40.40 | 118.95 | 228 | FM-12 |
| YU XIAN, CH | 39.83 | 114.57 | 910 | FM-12 |

[Figure]

Figure I: Figure 4 of the preprint manuscript with modified color palette.

adjusted. The current choice makes it difficult to distinguish between them. More contrasting colors would improve readability, especially since these sectors have significant regional influence.

Thank you for the suggestion. We have changed the color palette in Figure 4. The new figure is shown here as Fig. I

**References**

Guevara, M., Jorba, O., Tena, C., Denier van der Gon, H., Kuenen, J., Elguindi, N., Darras, S., Granier, C., & Pérez García-Pando, C. (2021). Copernicus Atmosphere Monitoring Service TEMPOral profiles (CAMS-TEMPO): Global and European emission temporal profile maps for atmospheric chemistry modelling. *Earth System Science Data*, *13*(2), 367–404. https://doi.org/10.5194/essd-13-367-2021

Maasakkers, J. D., Jacob, D. J., Sulprizio, M. P., Turner, A. J., Weitz, M., Wirth, T., Hight, C., DeFigueiredo, M., Desai, M., Schmeltz, R., Hockstad, L., Bloom, A. A., Bowman, K. W., Jeong, S., & Fischer, M. L.

(2016). Gridded National Inventory of U.S. Methane Emissions. *Environmental Science & Technology*, *50*(23), 13123–13133. https://doi.org/10.1021/acs.est.6b02878

Pisso, I., Sollum, E., Grythe, H., Kristiansen, N. I., Cassiani, M., Eckhardt, S., Arnold, D., Morton, D., Thompson, R. L., Groot Zwaaftink, C. D., Evangeliou, N., Sodemann, H., Haimberger, L., Henne, S., Brunner, D., Burkhart, J. F., Fouilloux, A., Brioude, J., Philipp, A., . . . Stohl, A. (2019). The Lagrangian particle dispersion model FLEXPART version 10.4. *Geoscientific Model Development*, *12*(12), 4955–4997. https://doi.org/10.5194/gmd-12-4955-2019

Schulzweida, U. (2020). *Climate data operators (CDO) user guide* (Version 2.3.0). 10.5281/zenodo.10020800

Yang, Y., Zhou, M., Langerock, B., Sha, M. K., Hermans, C., Wang, T., Ji, D., Vigouroux, C., Kumps, N., Wang, G., De Mazière, M., & Wang, P. (2020). New ground-based Fourier-transform near-infrared solar absorption measurements of $XCO_2$, $XCH_4$ and XCO at Xianghe, China. *Earth System Science Data*, *12*(3), 1679–1696. https://doi.org/10.5194/essd-12-1679-2020

Yang, Y., Zhou, M., Wang, T., Yao, B., Han, P., Ji, D., Zhou, W., Sun, Y., Wang, G., & Wang, P. (2021). Spatial and temporal variations of $CO_2$ mole fractions observed at Beijing, Xianghe, and Xinglong in North China. *Atmospheric Chemistry and Physics*, *21*(15), 11741–11757. https://doi.org/10.5194/acp-21-11741-2021